# Encoding Expert Knowledge into Federated Learning using Weak Supervision

## Abstract

Learning from on-device data has enabled intelligent mobile applications ranging from smart keyboards to apps that predict abnormal heartbeats. However, due to the sensitive and distributed nature of such data, it is onerous to acquire the expert annotations required to train traditional supervised machine learning pipelines. Consequently, existing federated learning techniques that learn from on-device data mostly rely on unsupervised approaches, and are unable to capture expert knowledge via data annotations. In this work, we explore how to codify this expert knowledge using programmatic weak supervision, a principled framework that leverages *labeling functions* (i.e., heuristic rules) in order to annotate vast quantities of data without direct access to the data itself. We introduce Weak Supervision Heuristics for Federated Learning (`WSHFL`[1]), a method that interactively mines and leverages labeling functions to annotate on-device data in cross-device federated settings. We conduct experiments across two data modalities: text and time-series, and demonstrate that `WSHFL` achieves competitive performance compared to fully supervised baselines without the need for direct data annotations. Our code is available at https://anonymous.4open.science/r/wshfl_pipeline-A13C/

## 1 Introduction

Learning from on-device data has the potential to enable increasingly intelligent mobile applications (McMahan et al., 2017), from smart keyboards that boost usability (Hard et al., 2018) to health apps that improve patient outcomes (Fitzpatrick et al., 2017; Bui & Liu, 2021). Nevertheless, on-device data cannot be annotated by external experts (Wang et al., 2021a): it is too large in scale to justify point-by-point labeling, and it is too sensitive to be transmitted and peeked at. Thus, previous efforts to train models on this type of data have mostly relied on unsupervised methods (Hard et al., 2018; Lu et al., 2021) or have used user-contextual signals as supervision (Yang et al., 2018). However, for some critical applications, these approaches fall short.

As a motivating example, consider training an arrhythmia detection model using electrocardiogram (ECG) data obtained with smart watches. This task requires both respecting the sensitive nature of the data and capturing clinicians' expertise, e.g., via annotations of the ECG waveforms. In this work, we consider federated learning to accomplish the former: keeping the data isolated on-device and, instead, exchanging model parameters (McMahan et al., 2017; Wang et al., 2021a). The question of how to capture the clinicians' expertise into the federated model, however, is an active area of research (Jeong et al., 2020; Liu et al., 2021; Zhuang et al., 2021a; Wu et al., 2021) that is central to our work.

In this paper, we explore a particular new strategy for codifying expert knowledge into cross-device federated models: using Labeling Functions (LFs), functions that assign potentially imperfect labels to subsets of the data and that can be used to automatically label training data (Ratner et al., 2017; Rühling Cachay et al., 2021). Encoding supervision through LFs is referred to as Programmatic Weak Supervision (PWS) (Ratner et al., 2016; Zhang et al., 2022), and it has had success in centralized settings (Fries et al., 2019; Dunnmon et al., 2020; Goswami et al., 2021; Dey et al., 2022). To the best of our knowledge, PWS has not been explored in federated scenarios, where the focus for

---

[1] pronounced as in wishful.

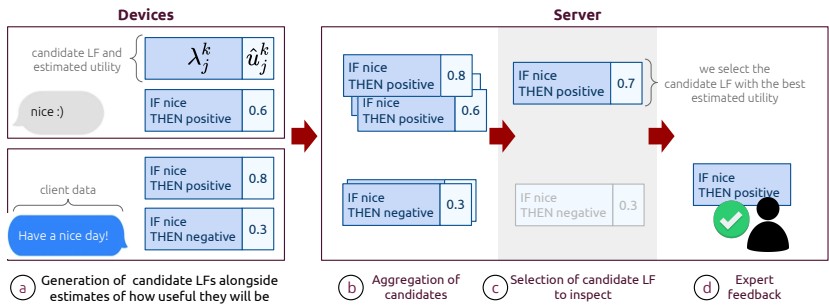

Figure 1: Visualization of WSHFL's strategy for generating LFs. Using on-device data, (a) candidate LFs $\lambda$ are generated alongside an estimate $\hat{u}$ of how probable an expert would find them useful. These candidates and estimates are then sent over to the server, where they are (b) aggregated before (c) one candidate is selected to be inspected by an expert. This (d) expert feedback is then used to generate future estimates $\hat{u}$.

encoding expert supervision has been on semi-supervised and self-supervised approaches for image and text data (Jeong et al., 2020; Liu et al., 2021; Zhuang et al., 2021a; Wu et al., 2021).

We introduce Weak Supervision Heuristics for Federated Learning (WSHFL), a method for mining and leveraging LFs in a cross-device federated setting. WSHFL proceeds in two stages: (i) the mining of LFs (or heuristics) and (ii) the training of the PWS model. In the first step, illustrated in Figure 1, WSHFL automates the crafting of LFs (Varma & Ré, 2018; Boecking et al., 2020), incorporating expert feedback on the generated LFs they consider useful. In this step, only parameterized LFs are exchanged, while the data is kept isolated on-device. In the second step, WSHFL trains a PWS model given the LFs from the previous step (Rühling Cachay et al., 2021).

In particular, we argue that the main challenge of adopting PWS into cross-device federated methods is the crafting of LFs. In practice, crafting these functions is a data-dependent process, as experts rely on available validation data in order to extract and assess dataset-specific heuristics (Varma & Ré, 2018; Boecking et al., 2020; Zhang et al., 2022). In federated learning, however, experts cannot freely explore the on-device data. To tackle this obstacle, we automatically generate LFs based on the distributed data and expert feedback at the central server (Boecking et al., 2020).

The key contributions of our work are as follows:

1. We introduce PWS into the federated setting, with the objective of encoding experts' knowledge into federated models through their inspection of candidate LFs that are mined from the on-device data. To this end, we propose approaches for two components of a standard PWS workflow for the federated set-up: the generation of candidate LFs, and the training of a model given LFs selected by the expert (Zhang et al., 2022).
2. We conduct experiments on three datasets across two data modalities, text and time-series, demonstrating the feasibility of the proposed approach compared to a fully supervised baseline. We also investigate each of its components, demonstrating their independent utility.
3. Our work is amongst the first to learn classification models from unlabeled distributed time-series data. Previous similar work has assumed access to labels (Zhang et al., 2020; Xu et al., 2021; Choudhury et al., 2019), while we only consider expert supervision over LFs.

## 2 RELATED WORK

**Programmatic Weak Supervision.** Programmatic weak supervision (PWS) has been proposed as an alternative to the expensive and time-consuming process of point-by-point labeling used for supervised machine learning. PWS leverages multiple sources of potentially noisy supervision, expressed as LFs, to label large quantities of data (Zhang et al., 2022). LFs, such as the one presented in Fig. 2, can be imperfect and may generate mutually conflicting labels on certain data points. Thus, a *label model* (Ratner et al., 2016; Rühling Cachay et al., 2021) is used to aggregate the noisy votes of labeling functions into training labels. These labels are then used to train an *end model*, which

learns to generalize the relationship between features and the estimated labels. Recent studies have also explored end-to-end approaches that couple the label and end models, leading to remarkable performance (Rühling Cachay et al., 2021). To the best of our knowledge, the PWS literature has only focused on centralized settings so far.

**Automatic Mining of LFs.** Existing methods have aimed to automate the creation of LFs given some extra supervision such as seed LFs (Li et al., 2021), labeled data (Varma & Ré, 2018; Awasthi et al., 2020), class descriptors (Gao et al., 2022), or instance-wise expert feedback (Nashaat et al., 2020). Boecking et al. (2020) introduced an algorithm that learns useful heuristics from user feedback at the LF level. WSHFL leverages this particular type of expert supervision while tackling challenges inherent to the federated scenario.

```python
def nice_lf(review):
    if "nice" in review:
        return POSITIVE
    else:
        return ABSTAIN
```

Figure 2: Example of a labeling function. If the unigram "nice" appears in a review, then it votes for the positive class, otherwise it abstains from voting.

**Semi-supervised and Self-supervised methods in Federated Learning.** One can also codify expert knowledge into federated models by using self-supervised or semi-supervised learning. Recent works that have studied these alternatives rely on a centralized dataset available for annotation by the experts, and augment the federated learning procedure with techniques such as consistency regularization (Jeong et al., 2020; Liu et al., 2021) and contrastive learning (Zhuang et al., 2021a; Wu et al., 2021). However, these techniques usually depend on the ability to augment their data at scale, and have thus been mostly used on image data (Jeong et al., 2020; Zhuang et al., 2021a; Wu et al., 2021; Liu et al., 2021). We provide an deeper discussion of these methods and their limitations in Appendix A.7 .

**Time-series federated learning with expert supervision.** Federated learning from time-series data is an active area of research (Ding et al., 2022). Nevertheless, prior work on federated learning with time-series is limited to semi-supervised or unsupervised problems such as anomaly detection (Liu et al., 2020; Huong et al., 2021), regression (Brophy et al., 2021) and forecasting (Tonellotto et al., 2021). While some studies have considered supervised classification in a cross-silo setting, they assume access to labels with a primary emphasis on privacy preservation (Zhang et al., 2020; Xu et al., 2021; Choudhury et al., 2019).

## 3  WEAK SUPERVISION HEURISTICS FOR FEDERATED LEARNING

### 3.1  PROBLEM FORMULATION

We aim to train an end model $f$ from unlabeled data distributed across devices or clients These clients communicate with a server that has no access to the clients' data and orchestrates training. We assume stateless clients as is the norm in cross-device federated learning. For each client $k$, let $(x^k, y^k) \sim \mathcal{D}^k, \mathcal{D}^k \sim \mathcal{P}$ be the data generating distribution, where $x_i^k \in \mathcal{X} = \mathbb{R}^d$ and the $y^k \in \mathcal{Y} = \{1, \ldots, C\}$. As is common in the federated setting, we assume the data between clients is not identically distributed, but all clients share the same feature and label space, *i.e.* $\forall k, \; x_i^k \in \mathcal{X}, y^k \in \mathcal{Y}$. Each client only observes a sample $X_k = \{x_i^k\}_{i=1}^{n_k}$ of $n_k$ unlabeled data points. We also have access to an expert located at the server who is able to determine the utility of a given LF. In Section 3.2, we formalize a notion of utility.

Our ultimate goal is to collaboratively train an end model $f : \mathcal{X} \to \mathcal{Y}$. To this end, WSHFL first uses the distributed data to generate candidate LFs $\lambda = \lambda(x) \in \{0\} \cup \mathcal{Y}$, where $0$ means that the LF abstained from labeling any class. Then, WSHFL identifies a set of useful LFs $\mathcal{L}^*$ based on the expert's feedback (Boecking et al., 2020). Finally, WSHFL uses $\mathcal{L}^*$ to train a PWS model on the clients' data, obtaining the resulting end model $f$.

### 3.2  AUTOMATIC MINING OF LFs

In this step, WSHFL sequentially shows candidate LFs to the expert at the server. In each step $t$, the expert inspects a given candidate $\lambda_t$ and assigns it an expert label $u_t \in \{0, 1\}$ corresponding to whether they believe its accuracy $\alpha_t = P(\lambda_t(x) = y | \lambda_t(x) \neq 0)$ is better than random. This step finally returns those LFs that the expert believed were accurate: $\mathcal{L}^* = \{\lambda_j \in Q_T : u_j = 1\}$.

Algorithm 1 describes our general procedure. The main challenges to highlight are (1) the generation of candidates LFs in a federated scenario (lines 18 and 13), and (2) the selection of the LFs we show to the expert (line 14).

GENERATION OF CANDIDATE LFS

To generate candidate LFs in a federated setting, WSHFL leverages two domain specific processes: (i) A client process that takes the unlabeled data $\{x_i^k\}_{i=1}^{n_k}$ and produces candidate heuristics $\mathcal{L}_k = \{\lambda_j^k\}_{j=1}^{p_k}$ in each individual client (see line 18 in Algorithm 1). And (ii) a server process that aggregates similar candidates proposed across clients into $G$ LFs $\mathcal{L}' = \{\lambda_j'\}_{j=1}^{G}$ (see line 13 in Algorithm 1).

A parameterization of the $\mathcal{L}_k$ generated at the clients is shared with the server. In Section 4, we describe this parameterization, as well propose different generation and aggregation methods for the two data modalities we work with.

SELECTION OF NEXT LF TO INSPECT

We cast this task as an *active search* problem (Boecking et al., 2020; Garnett et al., 2012), where we sequentially inspect the candidate LFs in order to discover those that the expert would assign expert label $u_j = 1$ . To do this , at time step $t$, we require access to the posterior probability $P(u = 1|\lambda, Q_{t-1})$, where $Q_{t-1} = \{(\lambda_j, u_j)\}_{j=1}^{t-1}$ corresponds to the previously inspected candidates and their expert labels.

To estimate this probability, WSHFL trains a model $h_k$ in each client that predicts $u_j$ given the client-specific representation $\tau_k(\lambda_j) = (\lambda_j(x_1^k), \ldots, \lambda_j(x_{n_k}^k))$, for all elements of $Q_{t-1}$. This model is then used to obtain estimates of $\hat{u}_j^k = h(\tau_k(\lambda_j^k))$ for the candidates that the client generates, which are shared with the server alongside the proposed candidates. We describe this in function TrainClient in Algorithm 1.

---

**Algorithm 1:** WSHFL mining of labeling functions

**Input:** Number of expert queries $T$, seeds $S$.

1   $Q_0 \leftarrow S$
2   **for** $t = 1, \ldots, T$ **do**
3     $\lambda_t \leftarrow$ FederatedAcquisition$(Q_{t-1})$
4     $u_t \leftarrow$ ExpertQuery$(\lambda_t)$
5     $Q_t \leftarrow Q_{t-1} \cup (\lambda_t, u_t)$

**Output:** $\{\lambda_j \in Q_T : u_j = 1\}$

6   **Function** FederatedAcquisition$(Q)$
7     $\mathcal{L}_0 \leftarrow \emptyset$
8     **for** $r = 1, \ldots, R$ **do**
9      Select $K$ clients at random.
10      **retrieve from each client**
11       $\mathcal{L}_k \leftarrow$ TrainClient$(Q)$
12     $\mathcal{L}_r = \mathcal{L}_{r-1} \cup \bigcup_{k=1}^{K} \mathcal{L}_k$
13     $\mathcal{L}' \leftarrow$ Aggregate$(\mathcal{L}_R)$.
14     $\lambda \leftarrow$ SelectBest$(\mathcal{L}')$.
15     **Return** $\lambda$

16   **Function** TrainClient$(Q)$
17     Train neural network $h_k : \tau_k(\lambda) \rightarrow u$ using $Q$.
18     Generate candidate LFs $\mathcal{L}_k = \{\lambda_j^k\}_{j=1}^{p_k}$.
19     Use $h_k$ to estimate $\hat{u}_j^k$ for $\lambda_j^k \in \mathcal{L}_k$.
20     **Return** $\{(\lambda_j^k, \hat{u}_j^k)\}_{j=1}^{p_k}$

---

When WSHFL aggregates similar candidates at the server into $\lambda' \in \mathcal{L}'$, it also aggregates their accuracy estimates $\hat{u}_j$, treating them as sample estimates of $P(u' = 1|\lambda', Q_{t-1})$. More concretely, let $A$ be the collection of candidates being aggregated, to estimate our posterior probability, we use a $1 - \delta$ lower confidence bound on the mean

$$P(u' = 1|\lambda', Q_{t-1}) = \frac{1}{|A|} \sum_{j \in A} \hat{u}_j - \sqrt{\frac{\log(\frac{2}{\delta})}{2|A|}}.$$

We use a lower bound to account for the variance of the simple mean when collections of candidates have different support, i.e., $|A|$. Finally, we use a one-step look-ahead search strategy, picking the aggregate $\lambda' \in \mathcal{L}'$ with the highest $P(u' = 1|\lambda', Q_{t-1})$. Because of the way we constructed our posterior, we will pick an aggregate with high estimated mean accuracy and high support. This is especially relevant in scenarios with heterogeneous data distributions.

### 3.3 TRAINING OF THE PWS MODEL

Once we have $\mathcal{L}^*$, we can use these LFs to train label model $g$ and the resulting end model $f$ on the clients' unlabeled data. In this work, we leverage the Weakly Supervised End-to-end Learner (WeaSEL) proposed by Rühling Cachay et al. (2021), a state-of-the-art PWS model. Like most PWS models, WeaSEL was proposed for centralized data. However, it's architecture makes it amenable to be learned in a distributed setting.

The key idea of `WeaSEL` is to use a two-player cooperative game between two models with different views of the unobserved label through the lens of the features and the LF votes, minimizing a pair of objectives of the form

$$L_f(\mathcal{D}) = \mathbb{E}_{\mathcal{D}}[L(y_f, \texttt{stop-grad}(y_g))] \quad \text{and} \quad L_g(\mathcal{D}) = \mathbb{E}_{\mathcal{D}}[L(y_g, \texttt{stop-grad}(y_f))]$$

where $L$ is a noise-aware loss (*e.g.*, cross-entropy), $y_f = f(x)$ and $y_g = P(y|\lambda)$ are probabilistic labels generated by the label model $g$ that takes both features $x$, LF outputs $\lambda(x)$ and class balances $P(y)$ as input. To intuitively understand this game, first assume that the probabilistic labels supplied by $g$ are accurate. `WeaSEL` can then train end model $f$ to generalize the relationship between these labels and the features of the data. On the other hand, assume end model $f$ already provides accurate predictions for our data. These predictions can thus be used as supervision to train $g$ to output correct probabilistic labels. The `stop-grad` operation naturally encodes this interpretation *i.e.* each model treats the other's prediction as the target.

In this work, we train `WeaSEL` in a federated setting, where the objectives become

$$L_F = \mathbb{E}_{\mathcal{D}^k \sim \mathcal{P}}[L_f(\mathcal{D}^k)] \quad \text{and} \quad L_G = \mathbb{E}_{\mathcal{D}^k \sim \mathcal{P}}[L_g(\mathcal{D}^k)].$$

Because we have access to a finite number of clients, and a finite sample of examples per client, we use empirical risk minimization to solve for these objectives.

We exchange $f$, $g$ and $\mathcal{L}^*$ throughout training. We assume global class balances $P(y)$ to be known, as is frequent in related work (Boecking et al., 2020; Ratner et al., 2019; Fu et al., 2020; Chen et al., 2021). Other works in centralized settings have proposed ways of estimating this quantity from validation data or from LF responses (Ratner et al., 2019). We leave the problem of estimating $P(y)$ from federated data as a direction of future work, and explore the interaction between a global class balance $P(y)$ and local client balances $P_k(y)$ in Section 6.2.

### ASSUMPTIONS

`WSHFL` relies on the ability to generate candidate LFs of varying quality, for which we use domain-specific processes. Previous work in mining LFs has observed that this generation process is possible for several applications (Varma & Ré, 2018; Boecking et al., 2020). We also rely on the ability of experts to determine whether a given LF is accurate. Once again, prior work has shown that domain experts are able to exercise this judgment, either while providing feedback of this type (Boecking et al., 2020), or while crafting LFs from scratch (Goswami et al., 2021; Dey et al., 2022; Fries et al., 2019; Dunnmon et al., 2020).

In this work we assume that the parameterized LFs can be freely shared with the server and, after aggregation and inspection by the expert, with other clients. We also assume estimates $\hat{u}_j, j \in A$, for a given $A$ to be independent in order to construct our lower bound on the posterior $P(u' = 1|\lambda', Q)$. This independence will not hold, for example, if the distribution over clients $\mathcal{P}$ changes over time (Kairouz et al., 2021).

## 4 LABELING FUNCTION GENERATION

**Text LFs.** We propose LFs that assign a label if a unigram is present in the data point, e.g., Figure 2. Otherwise, the LF abstains. Previous studies have found unigrams to be excellent sources of weak supervision (Gao et al., 2022; Boecking et al., 2020). A client $k$ can automatically generate $\mathcal{L}_k$ from the cross product of the set of possible labels and the unigrams in its vocabulary within a document frequency range[2]. In the server, we aggregate candidates with the same unigram and label.

**Time-series LFs.** These LFs are fully-parameterized by $(\tau, d_\tau, l)$, where $\tau \in \mathbb{R}^d$ is a time-series template, $d_\tau \in \mathbb{R}$ is a distance threshold, and $l \in \mathcal{Y}$ denotes the label. Given these parameters, and a distance function $\mathbf{d} : \mathbb{R}^d \times \mathbb{R}^d \to \mathbb{R}$, and a probability threshold $p$, each time-series LF has the following functional form:

$$\lambda(x; \tau, d_\tau, l) = \begin{cases} l, & \mathcal{F}(d_\tau, \mathbf{d}(x, \tau)) \geq p \\ 0, & \text{otherwise} \end{cases}, \text{ where } \mathcal{F}(x, x_0) = \frac{1}{1 + \exp{-\{x - x_0\}}}.$$

---

[2]The document frequency of a unigram is defined as the fraction of documents which contain at least one occurrence of the unigram.

Intuitively, the more a time-series $x$ looks like template $\tau$, the higher the probability it has of being assigned label $l$. We present an example of our time-series LFs in Figure 3. In this study, we use normalized euclidean distance as the distance function $\mathbf{d}$ (Ding et al., 2008; Mueen et al., 2009).

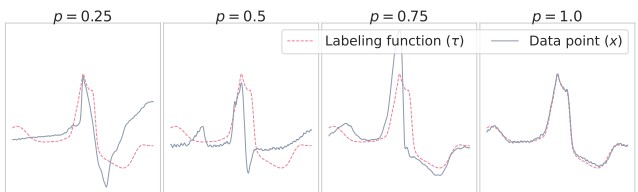

Figure 3: Example of a time-series labeling function representing an arrhythmia. We show 4 data points with increasing probabilities of belonging to the given class. These examples will be labeled as arrhythmias as we vary our probability threshold $p$.

We generate $\mathcal{L}_k$ by taking the cross-product of the set of possible labels and representative templates we find by clustering the data in each client. We define $d_\tau$ to be the distance of the cluster member farthest from the centroid. On the server, to aggregate these candidates, we cluster LFs with the same label from multiple clients. Each cluster then represents an aggregate LF: the cluster centroid serves as the template $\tau$, and the maximum $d_\tau$ serves as the new distance threshold.

## 5 EXPERIMENTAL SETUP

**Datasets.** For text, we use the Amazon product reviews dataset (Ni et al., 2019) and the IMDb movie reviews dataset (Maas et al., 2011). With both of these datasets we target a binary sentiment analysis task. In the Amazon dataset, we treat each unique reviewer as a different client, whereas on the IMDb dataset, we split reviews uniformly at random between clients. For time-series, we use the Massachusetts Institute of Technology – Beth Israel Hospital Arrhythmia Database (MIT BIH) (Moody & Mark, 2001; Goldberger et al., 2000). In this dataset, we solve a binary classification task of discriminating normal heart beats from arrhythmias, treating each patient as a different client. In Appendix A.4, we provide further details about our datasets.

**Methods and Models.** We featurize our text data using a pre-trained open-source sentence transformer (Reimers & Gurevych, 2019). For our arrhythmia detection task, we use the Modified Lead II from the raw ECG data sampled at 360Hz. Client models $h_k$ and label model $g$ are each two-layer perceptrons. For our text datasets, the end model $f$ is also a two-layer perceptron, while for our time-series data we use a one-dimensional CNN. We optimize our parameters on a validation dataset, and report the ROC AUC on a separate test dataset. We simulate an expert using an oracle that labels a LF as useful if it has a training accuracy of at least $0.7$. We provide further details about our models and hyper-parameters in Appendix A.4 and Appendix A.5. We also perform experiments with different expert thresholds, summarizing their results in Appendix A.2.

Unless mentioned otherwise, we repeat each experiment five times with different random seeds and report the mean and standard deviation.

**Baselines.** To the best of our knowledge, no prior work on federated learning has explored how to interactively encode expert supervision into on-device data. Hence, to evaluate `WSHFL`, we compare its predictive performance against 3 practical baselines. We also compare WSHFL with recent federated semi-supervised learning baselines with labels at the server on the IMDb dataset in Appendix A.7.

`Random:` First, we consider the scenario where the expert is shown randomly aggregated LFs, without considering their accuracy. This corresponds to changing line 14 in Algorithm 1 to $\lambda \leftarrow$ `SelectRandom`$(\mathcal{L}')$.

`Naive Greedy:` Next, we consider the setting where at each time-step $t$, the expert is shown the LFs with the highest $P(u' = 1|\lambda', Q_{t-1}) = \frac{1}{|A|} \sum_{j \in A} \hat{u}_j$. Simple mean estimates are natural in federated settings. However, we have to account for their variance due to our aggregating procedure.

`Supervised:` Finally, we compare `WSHFL` to the ideal setting where each device has access to ground truth labels and models are trained using FedAvg (McMahan et al., 2017). This baseline serves as an empirical upper bound for `WSHFL`'s performance.

# 6 RESULTS AND DISCUSSION

## 6.1 AUTOMATIC MINING OF LFs

Previous work on automatic mining of LFs has shown the importance of obtaining candidates with both high coverage and a high accuracy gap above chance (Boecking et al., 2020), where the coverage $l_j = P(\lambda_j(x) \neq 0)$ is the frequency at which $\lambda_j$ does not abstain. We plot these two quantities for the LFs inspected by the expert in Figure 8. Likewise, in Table 1, we present the percentage of LFs labeled as $u_j = 1$ out of those inspected, and their mean coverage. We observe how WSHFL promotes the mining of both high accuracy and high coverage heuristics across data modalities. Meanwhile, our greedy baseline fails to find high coverage LFs for our text dataset, successfully mining high coverage LFs only in our time-series experiments with the MIT BIH dataset.

To understand this behaviour, in Appendix A.3 we sketch the distribution of the proposed candidates' accuracies and coverages for our

| | $u_j = 1$ | |
|---|---|---|
| | **Percentage** | **Coverage** |
| | **Amazon** | |
| WSHFL | **30.13 +/- 5.27 %** | **1.94 +/- 2.28 %** |
| Greedy | 20.93 +/- 4.70 % | 0.04 +/- 0.04 % |
| Random | 9.20 +/- 3.55 % | 0.03 +/- 0.06 % |
| | **IMDb** | |
| WSHFL | 34.13 +/- 8.33 % | **3.47 +/- 3.40 %** |
| Greedy | **43.07 +/- 3.63 %** | 0.03 +/- 0.03 % |
| Random | 21.20 +/- 3.55 % | 0.07% +/- 0.37 % |
| | **MIT BIH** | |
| WSHFL | **97.80 +/- 1.40 %** | 2.41 +/- 3.87 % |
| Greedy | 96.40 +/- 2.50 % | **3.83 +/- 6.46 %** |
| Random | 44.20 +/- 6.60 % | 0.63 +/- 2.47 % |

Table 1: Percentage of LFs labeled as $u_j = 1$ out of those inspected by the expert, and their mean coverage. We can see how WSHFL mines both high accuracy and high coverage LFs for all of our datasets. In **bold**, the highest mean per dataset.

datasets. We observe how, for Amazon and IMDb, high accuracy candidates tend to have low coverage. Hence, the naive greedy baseline will end up with low coverage LFs. However, this is not the case for MIT BIH, where candidates with high accuracy also have good coverage. For this dataset, we expect greedy to be a competitive baseline.

In Figure 4, we use the mined LFs in a simple setting: a centralized majority vote label model. This is a simpler scenario than the PWS model we eventually want to train, which allows us to directly evaluate the quality of the mined LFs, and is a competitive baseline in the PWS literature (Rühling Cachay et al., 2021; Gao et al., 2022; Dey et al., 2022). We see how only WSHFL shows any meaningful improvement for our text datasets (Amazon and IMDb) while it performs comparably to our greedy baseline on the MIT BIH dataset.

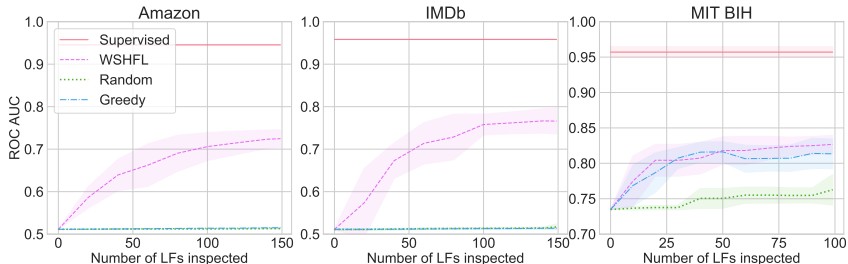

Figure 4: Results for a majority vote classifier given mined LFs. We observe how, as we present more LFs to the expert, WSHFL outperforms our baselines on our text datasets, and performs comparably to greedy on MIT BIH. Time-step 0 corresponds to an initialization as described in Appendix A.8.

## 6.2 TRAINING OF THE PWS MODEL

We validate that we can successfully train a WeaSEL model as proposed by Rühling Cachay et al. (2021) (Section 3.3) in a federated manner. For these experiments, we use a pre-curated set of LFs which we describe in Appendix A.6. In Figure 6, we show the results of our experiments. We present two baselines: the fully supervised baseline described above and an additional baseline corresponding to training WeaSEL in a centralized manner. With the latter, we aim to corroborate the utility of the used LFs in a previously studied setting of reduced complexity.

We study the behavior of federated training with three different algorithms: `FedAvg`, `FedProx`, and `FedAdam`. We observe how, for Amazon and IMDb, all three algorithms match the centralized performance after sufficient number of communication rounds. For the MIT BIH dataset, the best performing algorithm (`FedAvg`) achieves an ROC AUC of $82.66\%$ vs. $92.08\%$ of the centralized performance.

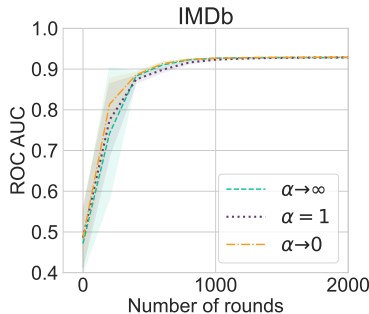

We also explore the effects of class imbalance on the performance of federated training. We use the method proposed by Hsu et al. (2019), which parameterizes the class distribution on a client by a vector $\mathbf{q} \sim \mathrm{Dir}(\alpha\mathbf{p})$ from a Dirichlet distribution, where $\mathbf{p}$ is a uniform prior and $\alpha > 0$ controls how much the class distributions across clients resemble each other. In Figure 5, we show results for training `WeaSEL` using `FedAvg` on the IMDb dataset. We observe how our results are consistent as we vary $\alpha$: when clients have identical class distributions ($\alpha \to \infty$), when clients have only one class each ($\alpha \to 0$), and for intermediate values. These results also suggest that it may be sufficient to specify the global class balance $P(y)$ even when the clients' class balances $P_k(y)$ differ.

Figure 5: Results of training `WeaSEL` in a federated setting on the IMDb dataset, given a set of curated LFs. We achieve consistent results as we vary the class distributions in each client, from one class per client ($\alpha \to 0$) to balanced classes ($\alpha \to \infty$). We present the test ROC AUC of the end model vs. the number of rounds of federated training.

### 6.3 PUTTING IT ALL TOGETHER

Finally, we demonstrate that we can (1) automatically generate useful LFs and (2) use them to successfully train a federated PWS model. Notice that this is a setting with higher complexity than the one in Figure 4. We show our results in Figure 7, training the PWS model using `FedAvg`. In our text datasets (Amazon and IMDb), we see how `WSHFL` is both more effective and efficient than our baselines at leveraging the expert's supervision. Meanwhile, for the MIT BIH dataset, it performs comparably to our greedy baseline. We also found WSHFL to outperform recent recent federated semi-supervised learning algorithms on the IMDb dataset (Fig.16).

## 7 CONCLUSIONS AND FUTURE WORK

Our work encodes expert supervision into on-device data in a scalable and distributed manner, extending the benefits of PWS to cross-device federated learning. We demonstrate how to train competitive federated models based on expert feedback at the LF level, avoiding both data-sharing and point-by-point labeling. Moreover, we investigate time-series modeling in federated settings, a modality that has been under-explored despite its prevalence in high-stakes scenarios.

**Societal Impact.** As with most federated learning algorithms, `WSHFL` makes frequent exchanges between clients and a central server in the form of parameterized models and LFs. There exists the risk of private information leaking through these exchanges. Understanding and mitigating the harms of exchanging model parameters is an active area of study (McMahan & Ramage, 2017;

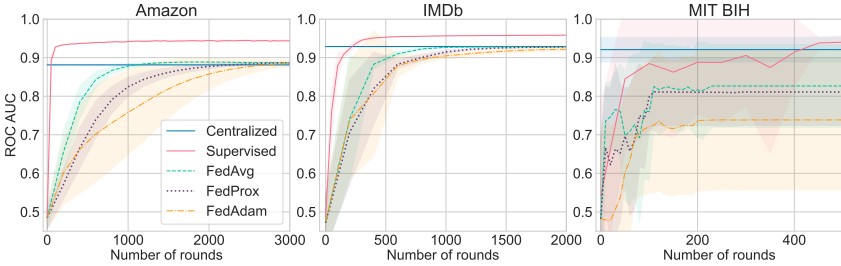

Figure 6: Results of training `WeaSEL` in a federated setting given a set of curated LFs. We observe that, given enough rounds of communication, we can match or come close to the performance of a centralized training scheme.

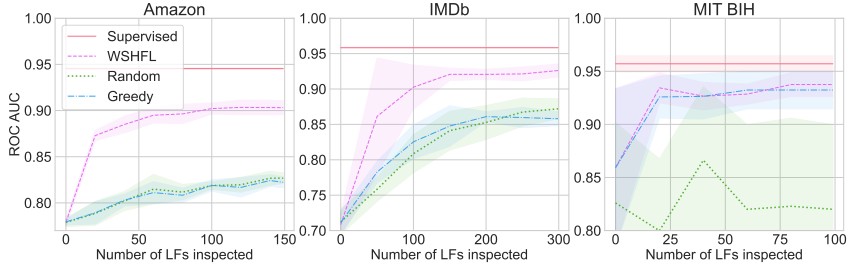

Figure 7: Results for WSHFL on our datasets. We observe how, as we present more LFs to the expert, WSHFL outperforms our baselines on our text datasets, and performs comparably to greedy on MIT BIH. Time-step 0 corresponds to an initialization as described in Appendix A.8.

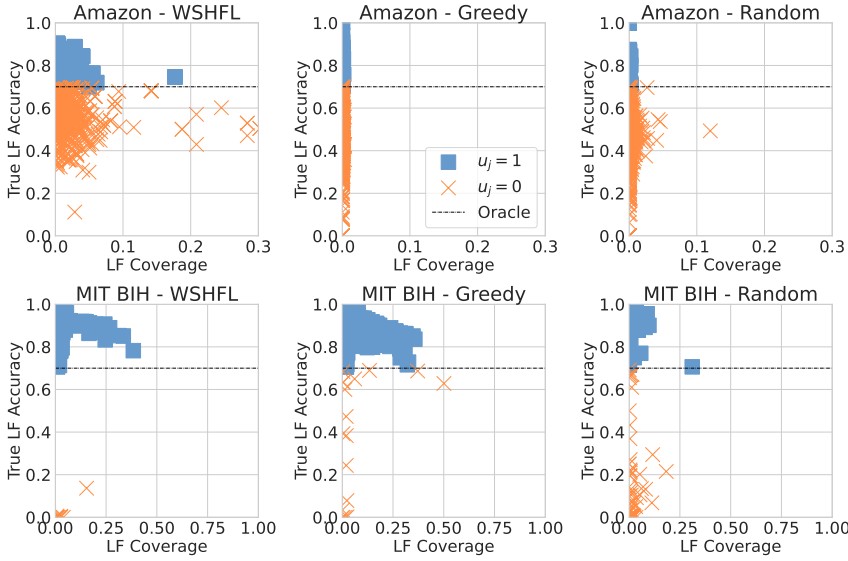

Figure 8: Training accuracies vs. coverages for all LFs inspected by the expert across five repetitions. We qualitatively observe how WSHFL promotes high accuracy and high coverage LFs across our three datasets. The black dotted line is the threshold at which our oracle starts labeling $u_j = 1$ in a LF, which we set to 0.7.

Bonawitz et al., 2022; Li et al., 2019). Depending on the LF parameterization, some LF families may adhere to privacy formalisms such as differential privacy, e.g., low-accuracy parametric classifiers. For other families, such as unigram LFs, these harms may not be fully understood and future work should mitigate them through techniques such as using a pre-set vocabulary (Chen et al., 2019).

**Studies in specific application domains.** To further establish the utility of the proposed approach, future work should study its performance and viability in applications and modalities beyond those explored in this work, e.g., clinical tabular or image data. A salient challenge in these studies will be the definition of LFs that can be easily inspected by experts. This research direction will also benefit from conducting user studies to evaluate the proposed LF generation mechanisms (Boecking et al., 2020). Immediate future work could conduct a study with clinical experts to evaluate our proposed time-series LFs.

**Improve selection of LFs.** We validate that WSHFL mines both accurate and high coverage LFs. However, future work could extend the active search formulation presented in this work using non-myopic strategies (Jiang et al., 2017) in the federated setting. Future studies could also equip this formulation with exploration capabilities, as is common in other sequential decision making settings (Sutton & Barto, 2018). Finally, we could conceive an active search formulation that is aligned with the performance of the end model itself, or with other properties of LFs, e.g., LF overlaps.

REPRODUCIBILITY STATEMENT

All our experiments were carried in a computing cluster with a typical machine having 128 AMD EPYC 7502 CPUs, 503 GB of RAM, and 8 NVIDIA 280 RTX A6000 GPUs. The code to reproduce our results will be open sourced upon acceptance. We have made an anonymous codebase available for review at `https://anonymous.4open.science/r/wshfl_pipeline-A13C/` To further aid reproducibility, we report the exhaustive set of hyper-parameters used in our experiments in Appendix. A.5. All datasets are also publicly available. The Amazon and IMDb datasets are can be downloaded from `https://nijianmo.github.io/amazon/index.html` and `https://huggingface.co/datasets/imdb`, respectively. The MIT-BIH dataset can be accessed from `https://www.physionet.org/content/mitdb/1.0.0/` and preprocessed using `https://github.com/physhik/ecg-mit-bih`. We

AUTHOR CONTRIBUTIONS

Omitted for blind review.

ACKNOWLEDGMENTS

Omitted for blind review.

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

# A    APPENDIX

## A.1    WSHFL OVERVIEW

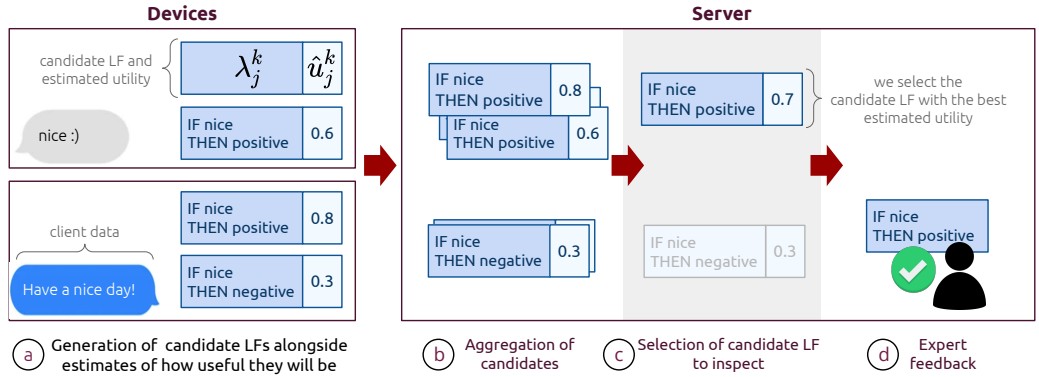

Figure 9: Visualization of WSHFL's strategy for generating LFs. Using on-device data, **(a)** candidate LFs $\lambda$ are generated alongside an estimate $\hat{u}$ of how probable an expert would find them useful. To generate candidate LFs, WSHFL leverages domain specific processes introduced in Sec. 4. Examples of generated text and time-series LFs are presented in App. A.10. The probability that a LF $\lambda$ will be found useful by the expert is estimated using a neural network $h_k$ in each client. $h_k$ leverages client-specific representations of $\lambda$, $\tau(\lambda)$, and the set of previously adjudicated LFs $Q_{t-1} = \{(\lambda_j, u_j)\}_{j=1}^{t-1}$ to estimate $\hat{u}$. These candidates and estimates are then sent over to the server, where similar candidate LFs are **(b)** aggregated using modality-specific processes described in Sec. 4. For e.g., text LF candidates with the same unigram and label are aggregated. Next, the server estimates the probability that an aggregated candidate LF will be found useful by the expert, and (c) one candidate is selected for their inspection. The (d) expert assigns an expert label $u \in \{0, 1\}$ corresponding to whether they believe its accuracy is better than random, and this feedback is then used to generate future estimates $\hat{u}$.

## A.2    ADDITIONAL RESULTS AND ABLATIONS

We investigate the behavior of our end-to-end WSHFL experiments with experts of different confidence levels. To do this, we vary the threshold at which our oracle labels a LF with $u = 1$. In Figure 10, we observe how WSHFL is robust to experts with different confidence levels, starting to degrade for Amazon and IMDb once the expert starts accepting LFs with accuracies close to random.

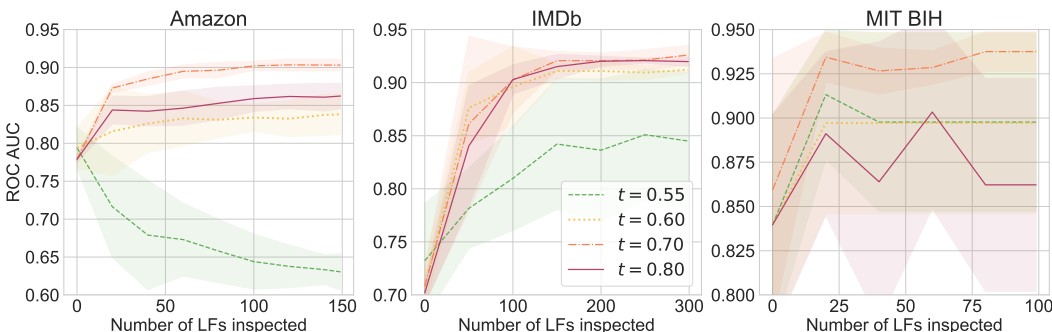

Figure 10: Results of our ablation experiments on WSHFL, where we vary the threshold of the oracle we use as an expert. We observe how our experiments are robust to a range of thresholds, yet may start to degrade when experts accept LFs too close to random.

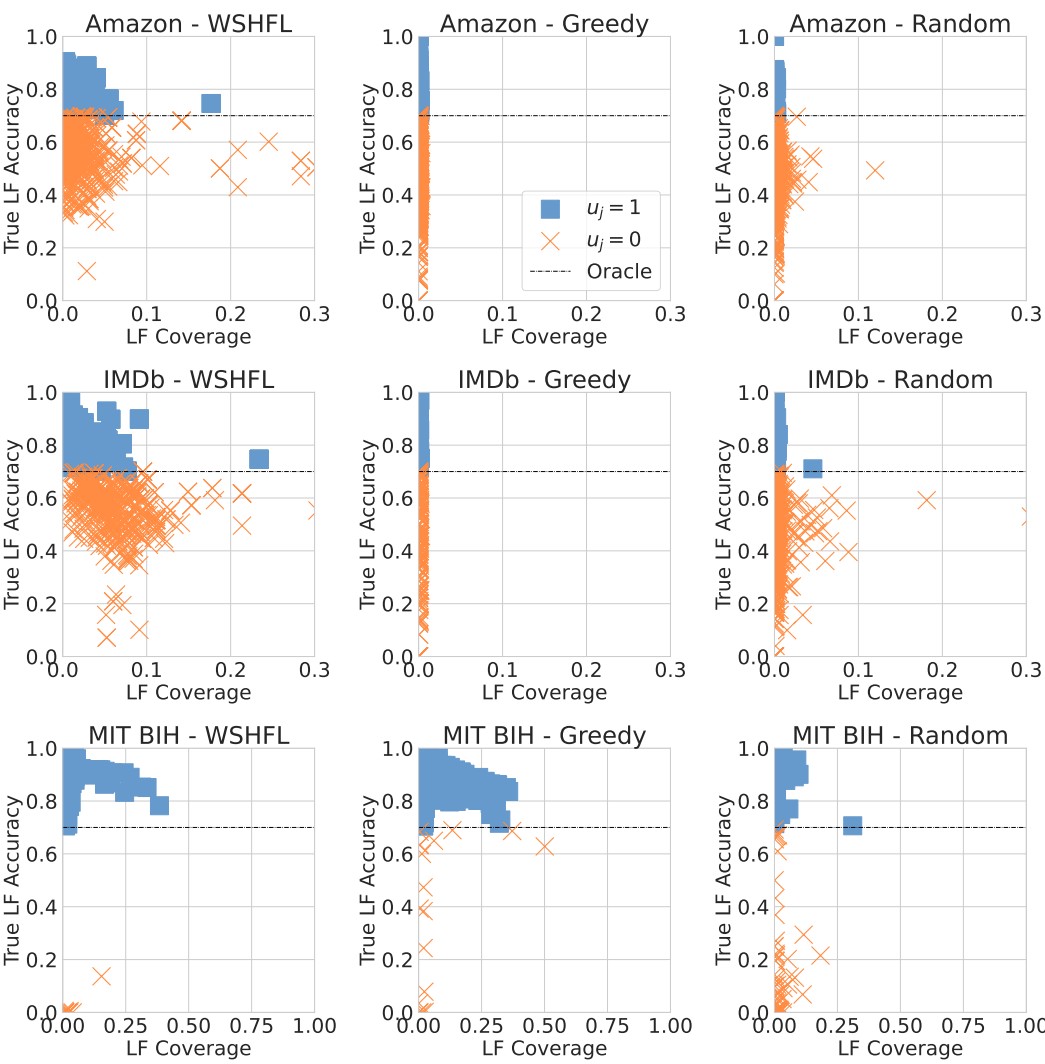

Figure 11: Training accuracies vs. coverages for all LFs inspected by the expert across five repetitions. We qualitatively observe how WSHFL promotes high accuracy and high coverage LFs across our three datasets. The black dotted line is the threshold at which our oracle starts labeling $u_j = 1$ in a LF, which we set to 0.7.

### A.3 Proposed Candidates Distribution

In Figure 13, we plot the accuracies and coverages for all the aggregated candidates at the server. This illustrates the distribution of LFs over which our method and baselines are sampling over. For all datasets, we intuitively observe a bimodal distribution of LFs based on their accuracy. This is because we exhaustively assign all classes to keyword/templates to generate LFs, hence for every accurate LF, we also have an equally inaccurate LF candidate. However, we found that the distribution of the accuracy of candidate LFs is different for text and time-series datasets. In particular, time-series LF candidates either had high or low accuracy, with few intermediate values.

### A.4 Datasets and Models

We provide a description of the datasets and models used in our work. We use federated versions of three different datasets: the Amazon product reviews dataset (Ni et al., 2019), the IMDb movie reviews dataset (Maas et al., 2011) and the Massachusetts Institute of Technology – Beth Israel Hospital Arrhythmia Database (MIT BIH) dataset (Moody & Mark, 2001; Goldberger et al., 2000). Statistics on the number of clients and examples in the different splits of these datasets are given in Table 2.

| | Num. Examples | Num. Clients | Mean Examples per Client (std) | Fraction of Positive Class |
|---|---|---|---|---|
| | **Amazon** | | | |
| Train | 119,725 | 738 | 162.22 (73.36) | 0.54 |
| Val | 20,090 | 123 | - | 0.54 |
| Test | 60,366 | 369 | - | 0.55 |
| | **IMDb** | | | |
| Train | 20,000 | 1000 | 20.0 (0.0) | 0.50 |
| Val | 5,000 | - | - | 0.49 |
| Test | 25,000 | - | - | 0.50 |
| | **MIT BIH** | | | |
| Train | 21,008 | 36 | 583.55 (461.72) | 0.58 |
| Val | 2,939 | 4 | - | 0.72 |
| Test | 4,153 | 8 | - | 0.60 |

Table 2: Details for datasets and partitions used in our experiments. We treat the validation and test partition as if it were centralized in the server.

#### A.4.1 Amazon

We use a subset of the Amazon product reviews dataset (Ni et al., 2019), solving a binary sentiment classification task. To construct our federated dataset, we first aggregate all categories with more than $100k$, and constructed clients $k$ based on the available reviewer ids. We then sampled reviewers in ascending order based on quantity $|P_k(y = 1) - 0.5|$ until we had at least $200k$ reviews. Intuitively, we looked for reviewers with class balances close to $0.5$. Finally, we performed a $60/10/30$ train/val/test split. We featurize this data using a pre-trained open-source sentence transformer (Reimers & Gurevych, 2019; Sentence Transformers, 2019), which outputs a feature vector of 768 dimensions. Our end model is a multilayer perceptron with two hidden layers of size 20 and RELU activations.

#### A.4.2 IMDb

We use the IMDb movie reviews dataset (Maas et al., 2011). This dataset has $25k$ training examples and $25k$ test examples. We further split the training set into $20k$ examples for training and $5k$ examples for validation, and create $1k$ training clients by splitting the reviews in the training set uniformly at random. We use the same featurization and end model as for the Amazon dataset.

Finally, in Figure 14, we show the distribution of local class balances $P_k(y)$ as we vary the parameter $alpha$ (see Section 6.2).

### A.4.3  MIT BIH

The MIT BIH dataset comprises of 48 half-hour excerpts of two-lead ambulatory ECG recordings from 47 subjects. The dataset contains beat-level annotations for a wide range of heart beats, ranging from normal to arrhythmia (e.g., left bundle branch block, premature ventricular contraction, etc). Since detecting all varieties of arrhythmia is challenging and not the primary goal of our study, we solved a simpler binary classification task of discriminating normal heart beats from arrhythmias.

On the MIT BIH dataset, we treat each patient as a different client. The patients used in each of the partitions in Table 2 are as follows:

| | |
|---:|:---|
| Validation: | 102, 115, 123, 202 |
| Test: | 101, 105, 114, 118, 124, 201, 210, 217 |
| Train: | All other patients |

In Figure 15, we illustrate the parameterization of the LFs we use for this dataset. As an end model, we train a one-dimensional convolutional neural network. Figure 21 shows the definition of the model we use. As input into our model, we use the Modified Limb lead II (MLII) obtained by placing electrodes on the chest, as is done in prior work (Goswami et al., 2021). We output a prediction for each window of 256 samples (sampled at 360Hz) around peaks given by previous preprocessing. Finally, we use early stopping based on our validation ROC AUC to avoid overfitting when training `WeaSEL` (Figure 6, Figure 7) and as the expert inspects the LFs (Figure 7).

### A.4.4  ADDITIONAL MODELS

Our label model is always a multi-layer perceptron with two hidden layers of size 20 and ReLU activations. When training this model, we set the class balance $P(y)$ to 0.5. Models $h_k : \tau_k(\lambda) \to u$ are multilayer perceptrons with two hidden layers of size 10 and ReLU activations.

### A.5  EXPERIMENT HYPERPARAMETERS

We describe the hyperparameters used for our experiments in Section 6. Section A.5.1 presents the parameters used in the experiments presented in Table 1, Figure 4, Figure 7 and Figure 11. Meanwhile, Section A.5.2 presents the parameters used in Figure 6 and Figure 7.

### A.5.1  AUTOMATIC MINING OF LFS

When generating time-series LFs as described in Section 4, we use the following parameters:

| | |
|---:|:---|
| Probability threshold ($p$) : | 0.2 |
| Number of clusters in client $k$ : | 3 `if` $n_k < 20$ `else` $\lfloor n_k/5 \rfloor$ |
| Number of clusters in server : | 500 |

When running Algorithm 1 to mine LFs, we train model $h_k$ in each client $k$ with the Adam optimizer (Kingma & Ba, 2014), using early stopping on the training loss. The hyperparameters that we use for this algorithm are as follow:

| | |
|---:|:---|
| Delta ($\delta$) : | 0.05 |
| Clients per round ($K$) : | 10 |
| Batch size : | 64 |
| Maximum number of epochs : | 200 |
| Learning rate : | 1e−3 |
| Weight decay : | 1e−4 |

Due to the difference in the total number of clients between datasets, in `FederatedAcquisition`, we use a different number of rounds $R$ per data modality. For text, we set $R$ to 10, while for time-series we set it to 1.

A.5.2 TRAINING OF THE PWS MODEL

Throughout these experiments, we compare the performance of training a federated version of `WeaSEL` using `FedAvg`, `FedProx` and `FedAdam`. For `FedAvg` and `FedAdam`, the client optimizer is mini-batch SGD, while for `FedProx` it includes a proximal term with weighted by $\mu > 0$. For all algorithms, we tune the hyperparameters using random search, exploring 20 sets of parameters and choosing the set with the best ROC AUC on the validation dataset. We perform this tuning once using the pre-curated set of LFs presented in Appendix A.6.

The hyperparameters that we explore are the following:

$$
\begin{aligned}
\log_{10}(\text{client learning rate}) &: \quad \text{Unif}[-2, -1] \\
\text{Temperature of } \texttt{WeaSEL} \text{ model} &: \quad \text{Unif}[10, 25] \\
\text{Number of client epochs} &: \quad \text{Unif}\{1, 3, 5\} \\
\text{Client momentum} &: \quad 0.9 \\
\text{Server momentum} &: \quad 0.9 \\
\text{Batch size} &: \quad 64 \\
\text{Clients per round} &: \quad 10
\end{aligned}
$$

For `FedProx` we tune $\mu$ in $\text{Unif}\{1e-3, 1e-2, 1e-1, 1\}$. For `FedAvg` and `FedProx`, we explore a $\log_{10}(\text{server learning rate})$ in $\text{Unif}\{-2, -1, 0\}$. For `FedAdam`, we explore the same hyperparameter in the range $\text{Unif}[-5, -4]$ for Amazon and IMDb, and in the range $\text{Unif}\{-4, -3, -2\}$ for MIT BIH.

In Table 3 we present the hyperparameters chosen after performing random search over the grids presented in Appendix A.5.2.

|  | Client lr | Server lr | Temperature | Epochs | $\mu$ |
|---|---|---|---|---|---|
| **FedAvg** | | | | | |
| Amazon | 3.35e−2 | 0.01 | 14.37 | 5 | - |
| IMDb | 4.15e−2 | 1.00 | 24.75 | 1 | - |
| MIT BIH | 2.66e−2 | 0.10 | 18.20 | 3 | - |
| **FedProx** | | | | | |
| Amazon | 3.70e−2 | 0.01 | 24.54 | 3 | 0.01 |
| IMDb | 4.60e−2 | 0.10 | 16.38 | 3 | 0.01 |
| MIT BIH | 6.81e−2 | 1.00 | 23.64 | 3 | 1.00 |
| **FedAdam** | | | | | |
| Amazon | 8.12e−2 | 1.81e−5 | 19.94 | 3 | - |
| IMDb | 4.82e−2 | 4.07e−5 | 22.50 | 5 | - |
| MIT BIH | 2.66e−2 | 1.00e−3 | 18.20 | 3 | - |

Table 3: Hyperparameters chosen after performing random search over the grids presented in Appendix A.5.2.

A.5.3 BASELINES

We tune the hyperparameters for two baselines in our experiments: a fully supervised baseline that we train using `FedAvg`, and a centralized version of `WeaSEL`. For both baselines, we tune their hyperparameters using random search, exploring 10 sets of parameters and choosing the set with the best ROC AUC on the validation dataset. For the `WeaSEL` baseline, we perform this tuning using the pre-curated set of LFs presented in Appendix A.6.

For our supervised (`FedAvg`) baseline, the hyperparameters that we explore are the following:

For our centralized (`WeaSEL`) baseline, the hyperparameters that we explore are the following:

In Table 4 we present the hyperparameters chosen after performing random search over the grids presented in Appendix A.5.3.

$$
\begin{aligned}
\log_{10}(\text{client learning rate}) &: \quad \text{Unif}[-4, -2] \\
\log_{10}(\text{server learning rate}) &: \quad \text{Unif}\{-2, -1, 0\} \\
\text{Number of client epochs} &: \quad \text{Unif}\{1, 3, 5\} \\
\text{Client momentum} &: \quad 0.9 \\
\text{Server momentum} &: \quad 0.9 \\
\text{Batch size} &: \quad 64 \\
\text{Clients per round} &: \quad 10
\end{aligned}
$$

$$
\begin{aligned}
\log_{10}(\text{learning rate}) &: \quad \text{Unif}[-4, -3] \\
\text{Temperature of } \texttt{WeaSEL} \text{ model} &: \quad \text{Unif}[10, 25] \\
\text{Momentum} &: \quad 0.9 \\
\text{Number of epochs} &: \quad 200 \\
\text{Batch size} &: \quad 64
\end{aligned}
$$

**Supervised** (`FedAvg`)

|         | client lr | server lr | epochs |
|---------|-----------|-----------|--------|
| Amazon  | 5.26e−3   | 1.00      | 3      |
| IMDb    | 5.26e−3   | 1.00      | 3      |
| MIT BIH | 2.15e−3   | 1.00      | 1      |

**Centralized** (`WeaSEL`)

|         | lr       | temperature |
|---------|----------|-------------|
| Amazon  | 3.06e−4  | 18.89       |
| IMDb    | 8.75e−4  | 22.99       |
| MIT BIH | 8.75e−4  | 22.99       |

Table 4: Hyperparameters chosen after performing random search over the grids presented in Appendix A.5.3.

### A.6 LABELING FUNCTIONS USED FOR FEDERATED WEASEL

For our experiments in Section 6.2, we use a set of pre-curated LFs. For Amazon and IMDb, we use the LFs reported by Boecking et al. (2020), which correspond to examples of heuristics that real users found useful when asked for the same type of feedback as the one described in this work. Meanwhile, for MIT BIH, we adopted an automated procedure to identify these LFs: using the method proposed by Boecking et al. (2020) in a centralized setting, using the same oracle as our experiments. The LFs used are detailed below.

- **Amazon:**
  - Positive: amazing, awesome, beautiful, beautifully, best, captivating, comfy, compliments, delightful, durable, easy, excellent, expected, fantastic, favorite, gorgeous, great, interesting, love, loves, perfect, perfectly, pleasantly, stars, strong, value, wonderful.
  - Negative: awful, bad, beware, boring, crap, disappointing, garbage, horrible, joke, junk, mess, money, poor, poorly, refund, sent, terrible, unusable, useless, waste, wasted, worse, worthless, worst, yuck, zero.
- **IMDb:**
  - Positive: amazing, art, beautiful, beautifully, breathtaking, brilliant, captures, delight, delightful, enjoyed, excellent, masterpiece, fantastic, favorites, finest, flawless, intelligent, joy, light, perfect, perfection, refreshing, superb, superbly, terrific, underrated, wonderful, wonderfully.
  - Negative: atrocious, awful, bad, boring, crap, decent, dreck, dull, failed, horrible, lame, laughable, lousy, mistake, pointless, poor, reason, redeeming, ridiculous, stinker, stupid, terrible, unfunny, unwatchable, waste, worst.
- **MIT BIH:** Figure 12 shows the LFs we use for our MIT-BIH experiments.

## A.7 Comparisons against Semi-supervised Federated Learning

### A.7.1 Related Work

Previous studies on federated learning make an unrealistic assumption that clients have ground-truth labels for their data (Kairouz et al., 2021). This is impractical as clients may not have time, resources, or the expertise to label task-specific on-device data. Recently, some studies have explored the possibility of combining semi-supervised learning with federated learning to enable clients without ground-truth labels to collaboratively train models. Below, we summarize recent advances in federated semi- and self-supervised learning, and highlight how they cannot be directly applied to our problem setting.

**Labels-at-client setting.** Some studies on semi-supervised federated learning assume that clients have partially labeled data (Wang et al., 2021b; Fan et al., 2022; Lin et al., 2022; Li et al., 2023; Liang et al., 2022; Lin et al., 2021; Yan et al., 2023). In our problem setting, clients do not have the expertise to annotate their own data.

**Labels-at-server setting and self-supervised FL.** Most related to our work is the research on semi-supervised federated learning with labels at the server. Techniques that work in the former setting assume that clients posses unlabeled data and that the server has access to a limited amount of annotated data (Zhang et al., 2021a; Diao et al., 2022; Long et al., 2020; Liang et al., 2022; Zhang et al., 2021b). Self-supervised studies do not assume access to any labels, and aim to learn task-agnostic representations from unlabeled distributed data (Makhija et al., 2022; Zhuang et al., 2021b). These methods, however, still require labeled data for the subsequent fine-tuning process.

**Limitations of current work.** Existing studies in semi- and self-supervised federated learning are designed for and evaluated on image datasets, and rely on modality-specific strong and weak data augmentation techniques such RandAugment (Cubuk et al., 2020), image shifting, and flipping, etc. Finding similar data augmentation techniques for text and time-series, the two modalities that we consider in this work, remains an area of open exploration (Bayer et al., 2022; Yue et al., 2022).

### A.7.2 Comparisons

We compare `WSHFL` on the IMDb dataset against two semi-supervised algorithms for federated learning with labels at server: SemiFL (Diao et al., 2022) and FRDG (Zhang et al., 2021b). We modify these baselines and use the identity function instead of weak and strong augmentations, and use the implementation provided by Diao et al. (2022). We present our results in Figure 16, and observe how `WSHFL` is closer to our fully supervised baseline than both SemiFL and FRDG.

For the semi-supervised experiments, we used 300 labeled examples at the server. This is equivalent to the number of times we query the expert in our experiments in Figure 7 for IMDb[3]. All federated experiments in this setting use the same number of training clients (1000) and clients per round (10). For SemiFL and FRDG, we use the hyperparameters suggested in the implementation by Diao et al. (2022), but set the local epochs to 3 after initial experimentation.

## A.8 Labeling Function Seeds

In our experiments, we initialize Algorithm 1 at time-step $0$ with a set $S$ of seed LFs. We use four seeds, two for each class, with a training accuracy above $0.7$ and thus labeled with $u = 1$. Furthermore, for our text datasets, in each repetition, we randomly sample four additional seeds in hopes of discovering LFs marked with $u = 0$. We don't perform this random sampling for the time-series modality as it heavily increased the variance of the ROC AUC at time-step $0$.

The seeds that were used throughout the experiments were:

- **Amazon:**
  - Positive: adorable, thoughtful.
  - Negative: stereotypical, horrible.

---

[3]Notice that these annotations refer to different entities: while `WSHFL` annotates LFs, SemiFL and FRDG require annotated examples.

- **IMDb:**
  - Positive: adorable, witty.
  - Negative: stereotypical, hated.
- **MIT BIH:** Figure 17 shows LFs we use as seeds for our MIT-BIH experiments. These LFs were chosen by the authors from the pool in Appendix A.6 based on visual inspection.

## A.9   EXAMPLES OF INSPECTED LABELING FUNCTIONS

We show examples of LFs considered useful by our simulated oracle, i.e., labeled with $u = 1$, on our three datasets. For Amazon and IMDb, we show the five heuristics most frequently annotated as useful, and break ties at random. For MIT BIH, we show the most accurate LFs found useful by the oracle, and break ties by selecting candidates that are visually dissimilar. The list of example heuristics is presented below:

- **Amazon:**
  - `WSHFL`:
    * Positive: size, love, recommend, perfect, highly.
    * Negative: poor, worse, boring, total, worst.
  - Greedy:
    * Positive: pastiche, pointofview, preschool, leveling, elm.
    * Negative: inarticulate, purchases, catnip, dart, selfabsorbed.
  - Random:
    * Positive: guitars, cliffhanger, break, domicile, define.
    * Negative: capable, tend, whichfacilitate, itunes, notable.
- **IMDb:**
  - `WSHFL`:
    * Positive: shows, performance, perfect, fun, excellent.
    * Negative: money, annoying, badly, awful, lame.
  - Greedy:
    * Positive: wordplay, caps, adoree, accelerated, ardour.
    * Negative: cadet, appendage, accuses, blueprints, beanies.
  - Random:
    * Positive: poltergeist, conversation, brawny, damages, 30mins.
    * Negative: critiquing, approved, effortless, completely, banner.
- **MIT BIH:**
  - `WSHFL`: We plot three labeling functions inspected by the expert using WSHFL in Figure 18.
  - Greedy: We plot three labeling functions inspected by the expert using our greedy baseline in Figure 19.
  - Random: We plot three labeling functions inspected by the expert using our random baseline in Figure 20.

## A.10   EXAMPLES OF LABELING FUNCTION GENERATION

Section 4 describes how LFs are generated in each client. Below we provide examples of how labeling functions are generated in each client for our text and time-series datasets.

**Text LFs.** Each client first identifies a set of unigrams (i.e., words) within a certain document frequency range. For example, say the client identifies 2 unigrams in their vocabulary: [nice, bad]. To create labeling functions, each client takes the cross product of these unigrams and the set of possible labels. If the set of possible labels is [negative sentiment, positive sentiment], then the client generates 4 LFs: [nice → positive sentiment, nice → negative sentiment, bad → positive sentiment, bad → negative sentiment]. To find keywords, we use the `CountVectorizer`[4] implementation in `scikit-learn`. Fig. 2 shows a programmatic representation of unigram LFs.

---

[4]`https://scikit-learn.org/stable/modules/generated/sklearn.feature_extraction.text.CountVectorizer.html`

**Time-series LFs.** In each client, we first cluster time-series into $k$ clusters using $k$-means clustering and normalized euclidean distance (Ding et al., 2008; Mueen et al., 2009). We use the `TimeSeriesKMeans`[5] implementation of `tslearn` to find time-series clusters. We refer to the cluster representatives ($\tau$, or the mean time-series) of these clusters as representative templates (see Fig. 12 and the red time-series in Fig. 15). To construct labeling functions, we take the cross product of these representative templates and the set of possible labels. As shown in Fig. 3, the closer a time-series is to a representative template, the more likely it is to be assigned to the class corresponding to the representative template.

## A.11   TABLE OF SYMBOLS

| Symbols | Description |
|---:|---|
| $k$ | Index for client |
| $\mathcal{X} = \mathbb{R}^d$ | $d$-dimensional feature space of data captured by each client. |
| $\mathcal{Y} = \{1, \ldots, C\}$ | Label space for each client. Each data point belongs to one of $C$ classes. |
| $u \in \{0, 1\}$ | The utility of a LF. An expert assigns a LF $u = 1$ if they believe that its accuracy is better than random. |
| $f : \mathcal{X} \to \mathcal{Y}$ | Downstream classifier. We use an MLP for text data, and ConvNet for time-series data. |
| $\lambda = \lambda(x) \in \{0\} \cup \mathcal{Y}$ | Given data as input, a LF votes for a particular class or abstains from voting ($\lambda(x) = 0$) |
| $\mathcal{L}^*$ | A set of useful LFs for a given dataset . `WSHFL` identifies a set of useful LFs based on the expert feedback. |
| $t \in \{0, \ldots, T\}$ | Time-steps |
| $\alpha = P(\lambda(x) = y \mid \lambda(x) \neq 0)$ | Accuracy of the LF $\lambda$ |
| $Q_t = \{(\lambda_j, u_j)\}_{j=1}^t$ | The set of LFs inspected by the expert at the server along with their utility labels at time $t$ |
| $n_k$ | Number of data points in client $k$ |
| $\mathcal{L}_k = \{\lambda_j^k\}_{j=1}^{p_k}$ | A set of $p_k$ candidate LFs for client $k$ |
| $\mathcal{L}' = \{\lambda_j'\}_{j=1}^G$ | Result of aggregation similar candidates proposed across clients into $G$ LFs |
| $P(u = 1 \mid \lambda, Q_{t-1})$ | The probability that $\lambda$ is useful, given the set of LFs already inspected by client until time-step $t - 1$. |
| $\tau_k(\lambda) = (\lambda(x_1^k), \ldots, \lambda_j(x_{n_k}^k))$ | Client-specific representation of $\lambda_j$. The representation of a LF is a vector of its responses on each of its data point. |
| $\hat{u}_j^k = h(\tau_k(\lambda_j^k))$ | $h_k$ is a neural network, which given the representations of a LF, predicts whether it would be deemed useful by the expert on the server. |
| $g : \{\mathcal{L}\} \to \mathcal{Y}$ | Programmatic weak supervision label model. |

Table 5: List of symbols and equations used in Section 3.

---

[5]`https://tslearn.readthedocs.io/en/stable/gen_modules/clustering/`
`tslearn.clustering.TimeSeriesKMeans.html`

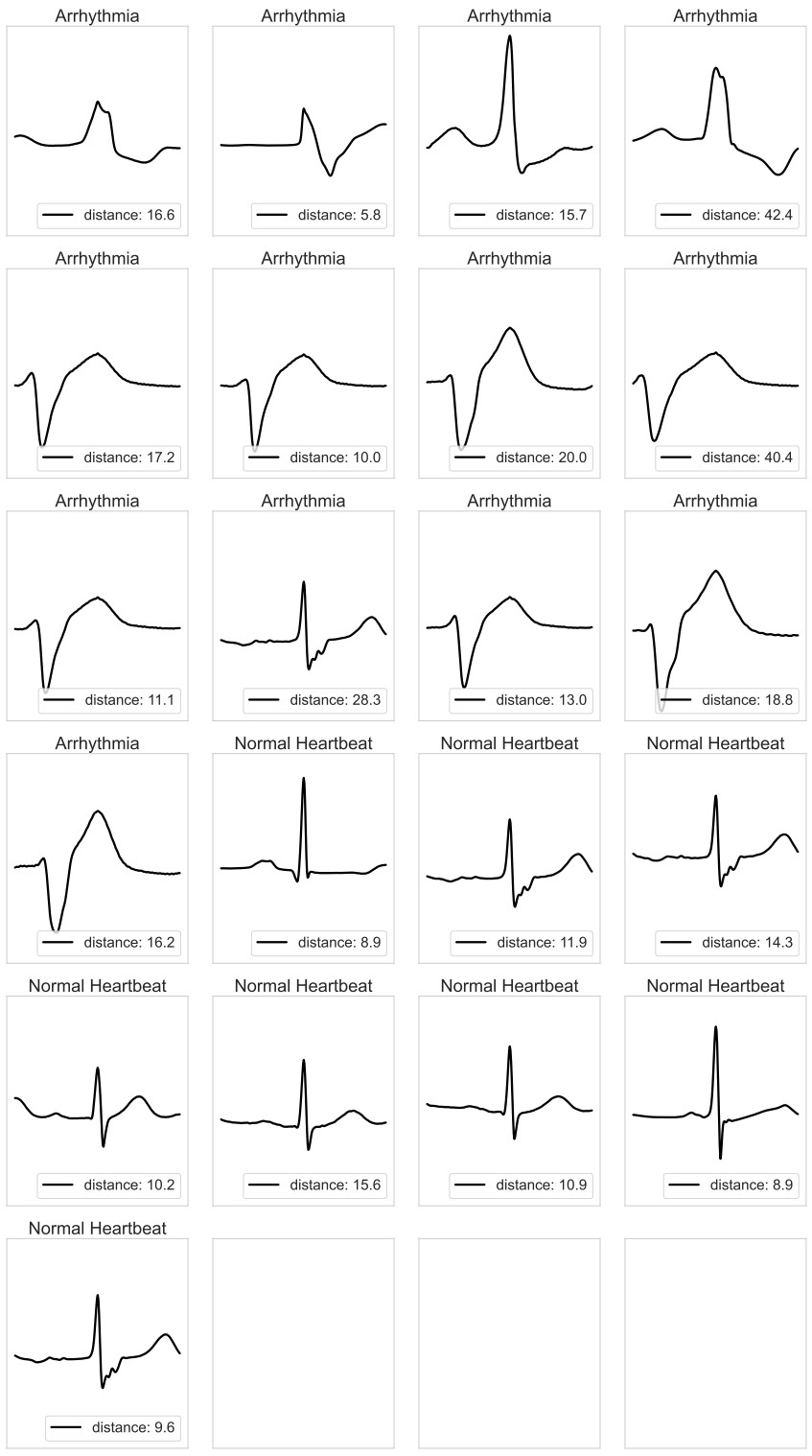

Figure 12: Visualization of the LFs used when training `WeaSEL` in a federated manner, using the MIT BIH dataset.

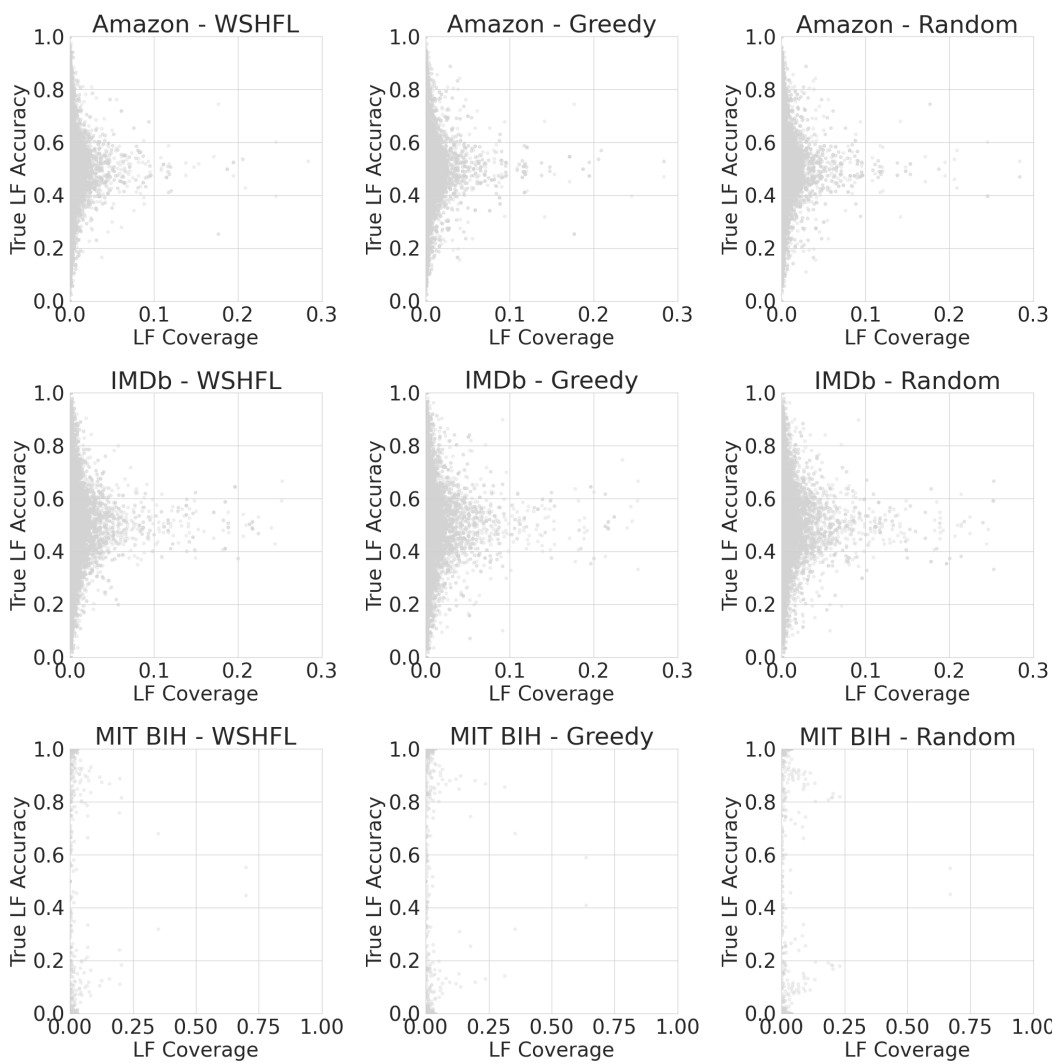

Figure 13: Training accuracies vs. coverages for the LFs aggregated at the server. We plot the candidates for the last time-step in one repetition chosen at random.

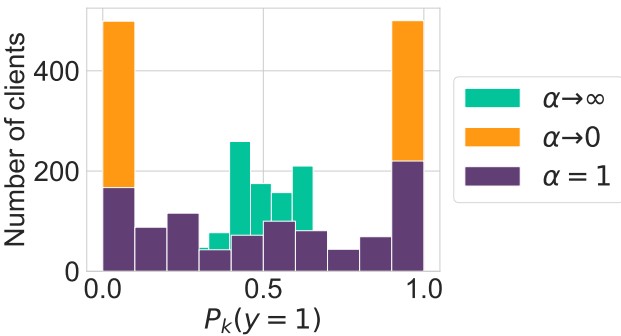

Figure 14: Histogram of local class balances $P_k(y)$ as we vary $\alpha$ for the IMDb dataset.

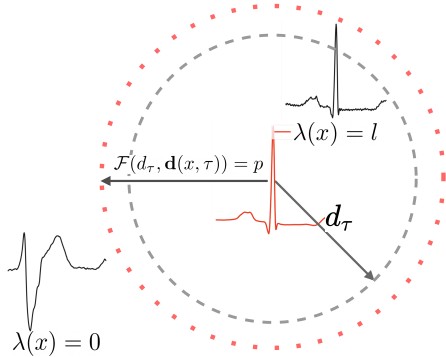

Figure 15: A time-series LF $\lambda$ is parameterized using a three-tuple $(\tau, d_\tau, l)$. The template $\tau$ shown in red is the centroid of a cluster, represented using the dashed lines $--$. $d_\tau$ is the radius of the cluster and corresponds to the distance of the cluster member farthest from $\tau$. Depending on the probability threshold $p$, this LF $\lambda$ will label data points $x$ as belonging to class $l$, or it will abstain from voting. This labeling threshold is denoted by the outer concentric circle $\cdots$.

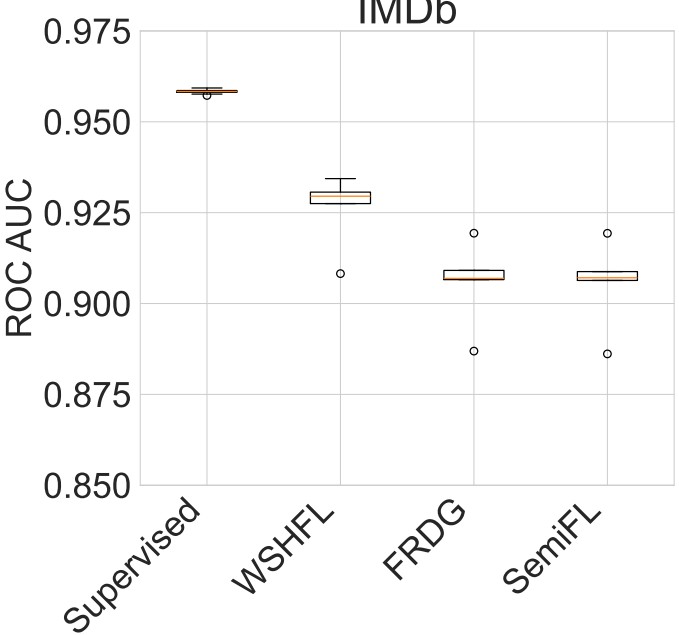

Figure 16: Comparisons against semi-supervised methods in federated learning, FRDG and SemiFL, for our IMDb dataset. We repeat each experiment five times and draw box plots with all repetitions. We observe how `WSHFL` outperforms the semi-supervised methods, and is closer in performance to the fully supervised baseline.

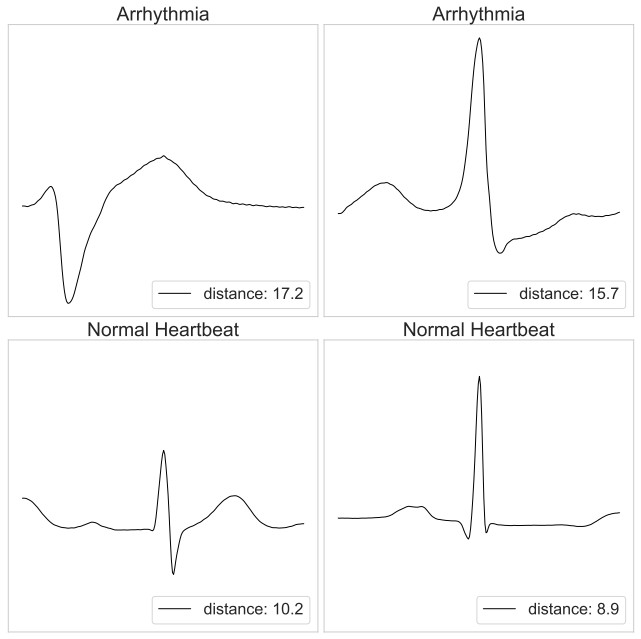

Figure 17: Visualization of the seeds used in our MIT BIH experiments. In our set up, arrhythmia corresponds to the positive class.

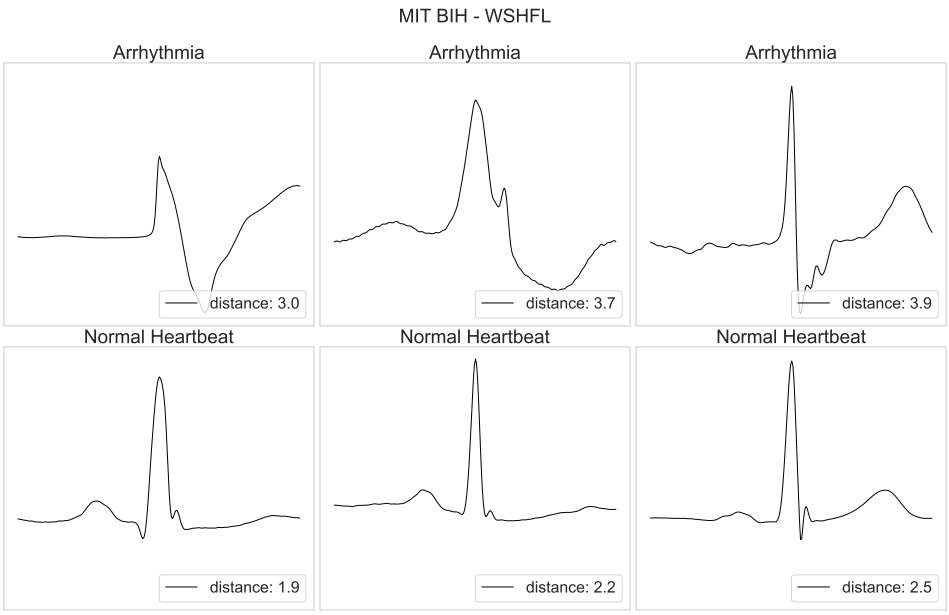

Figure 18: Visualization of the most accurate candidates inspected by the expert when using `WSHFL` with the MIT BIH dataset. We present the three top candidates per class and break ties by selecting candidates that are visually dissimilar.

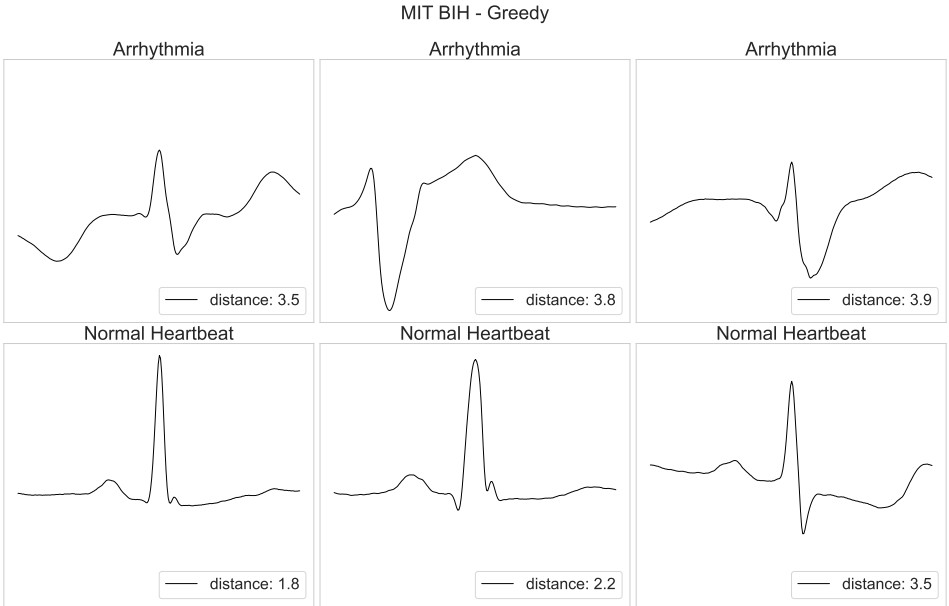

Figure 19: Visualization of the most accurate candidates inspected by the expert when using our greedy baseline with the MIT BIH dataset. We present the three top candidates per class and break ties by selecting candidates that are visually dissimilar.

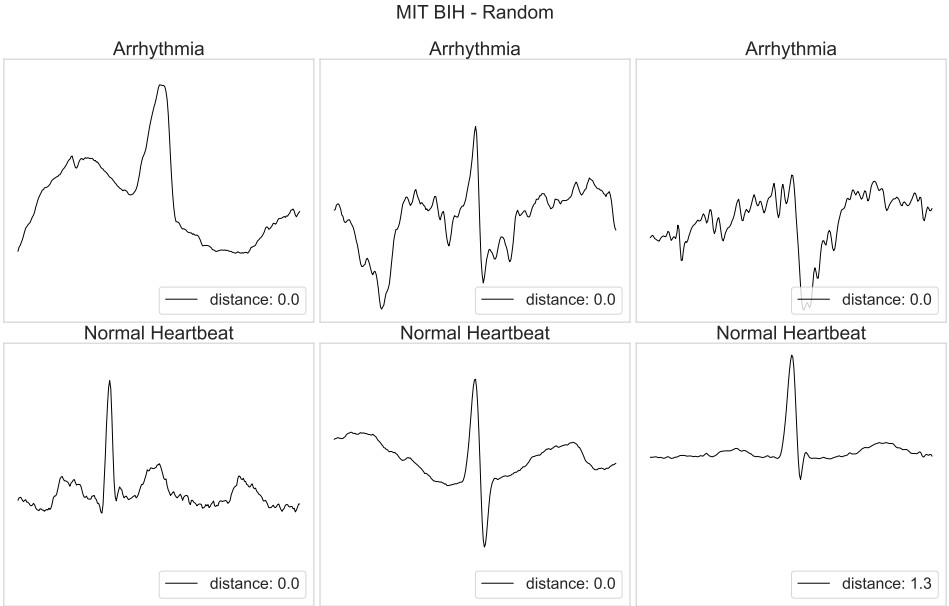

Figure 20: Visualization of the most accurate candidates inspected by the expert when using our random baseline with the MIT BIH dataset. We present the three top candidates per class and break ties by selecting candidates that are visually dissimilar.

```python
class ConvNet(nn.Module):
  def __init__(self, input_dim: int, output_dim: int,
               n_feature_maps: int = 4, name: str = 'convnet'):
    super().__init__()
    self.n_feature_maps = n_feature_maps
    self.output_dim = output_dim
    self.input_dim = input_dim
    self.name = name

    self.conv_block = nn.Sequential(*self._build_conv_modules())
    self.output_layer = nn.Sequential(*self._build_output_layer())

  def _build_conv_modules(self):
    return [
      nn.Conv1d(
        in_channels=self.input_dim,
        out_channels=self.n_feature_maps,
        kernel_size=8,
        stride=1,
        padding='same'),
      nn.BatchNorm1d(num_features=self.n_feature_maps),
      nn.ReLU(inplace=True),

      nn.Conv1d(
        in_channels=self.n_feature_maps,
        out_channels=self.n_feature_maps,
        kernel_size=5,
        stride=1,
        padding='same'),
      nn.BatchNorm1d(num_features=self.n_feature_maps),
      nn.ReLU(inplace=True),

      nn.Conv1d(
        in_channels=self.n_feature_maps,
        out_channels=self.n_feature_maps,
        kernel_size=3,
        stride=1,
        padding='same'),
      nn.BatchNorm1d(num_features=self.n_feature_maps),
      nn.ReLU(inplace=True),
    ]

  def _build_output_layer(self):
    return [
      nn.Linear(in_features=self.n_feature_maps, out_features=self.
                                      output_dim),
      nn.Softmax(dim=1)
    ]

  def forward(self, x, get_features=False):
    x = torch.unsqueeze(x, dim=1)
    output_conv_block = self.conv_block(x)

    output_avg_pool = nn.AvgPool1d(
      kernel_size=output_conv_block.shape[2],
      stride=1)(output_conv_block).squeeze()

    return self.output_layer(output_avg_pool)
```

Figure 21: Definition of the one-dimensional convolutional network used for the MIT BIH dataset.

