# OpenReview forum: "Encoding Expert Knowledge into Federated Learning using Weak Supervision"
_ICLR.cc/2024/Conference — Submitted to ICLR 2024_

### Official Review · Reviewer_qXVo · 2023-11-01

**Soundness:** 3 good
**Presentation:** 3 good
**Contribution:** 4 excellent
**Rating:** 6
**Confidence:** 4

**Summary:**

This paper proposed a novel paradigm, WSHFL,  for integrating programmatic weak supervision with on-device federated learning. The key ideas are composed of 2 components: automatic labeling function (heuristics) generations and federated learning with WeaSEL (Weakly Supervised End-to-end Learning). The authors conduct experiments to show that both components work across three benchmark tasks.

**Strengths:**

To the best of the reviewer’s knowledge, the idea of incorporating federated learning with programmatic weak supervision proposed in this paper is novel. This work is clearly motivated and it obviously fills the need for distributed programmatic weak supervision, especially with privacy concerns. I also appreciate the authors’ consideration of data beyond the text (i.e. ECG).

**Weaknesses:**

Overall, despite this paper(WSHFL)’s obvious merits, the method construction itself is incremental. The ideas of automated labeling functions generations have been a relatively well studied topic. If possible an open-source code base for federated learning with WeaSEL or other non end-to-end programmatic weak supervision would be very beneficial to the community.

**Questions:**

From the automatic labeling functions generation scheme, for example, the unigram proposal mechanism still can pose private data leakage risks. I would very much appreciate it if the authors can offer some ideas about initial solutions around this issue.

---

> ### Author Response · Authors · 2023-11-16
> **Thanks for reviewing! Our rebuttal**
>
> Dear Reviewer qXVo,
>
> We thank you for your time and effort in reviewing our paper.
>
> We are glad you find our work well motivated and novel, and that you recognize our efforts in performing experiments with ECG data. We have commented on the concerns about privacy raised by the reviewers in our *general response*. In that same response, we touch upon the concern that some aspects of the proposed method (automated LF generation) are already well studied in centralized settings.
>
> Regarding the suggestion to include an **open-source code base**, we plan to open-source our code when we release our paper. In the meantime, we have made the code anonymously available at https://anonymous.4open.science/r/wshfl_pipeline-A13C/README.md. We have included a link to this resource in the manuscript.
>
> Please let us know if there are any further areas we can improve on.

---

> > ### Author Response · Authors · 2023-11-21
> > **Summary of changes and a humble request for response**
> >
> > Dear Reviewer qXVo,
> > Thank you for your time and effort in improving our paper.
> >
> > We wanted to draw your attention to the numerous changes that we have made to the manuscript in response to your valuable comments and feedback. A summary of these changes is provided in the general comment we posted.  Based on your suggestions, we have made our code open source and provided a thorough discussion on the risk of privacy leakage from LFs. We have also outlined some initial solutions around the issues in this rebuttal and in the manuscript.
> >
> > *We are eagerly looking forward to your response. If our revisions have effectively addressed your concerns, we kindly request you to consider adjusting your scores accordingly.*
> >
> > Thank you once again for your thoughtful review.

---

### Official Review · Reviewer_W4Yv · 2023-11-01

**Soundness:** 2 fair
**Presentation:** 3 good
**Contribution:** 2 fair
**Rating:** 5
**Confidence:** 3

**Summary:**

The title is clear and gives a good indication of the paper's main focus. The abstract provides a concise overview of the problem, the proposed solution, and the results. The introduction sets the context well, highlighting the importance of learning from on-device data and the challenges associated with it. The motivating example of arrhythmia detection provides a real-world context, making it relatable for readers. This paper introduces a new method, WSHFL, that bridges the gap between federated learning and programmatic weak supervision, addressing a challenge in the field. Addressing the issue of on-device data annotation is timely and relevant, given the increasing importance of privacy and on-device computations.

**Strengths:**

The paper introduces the concept of Programmatic Weak Supervision (PWS) into the federated setting, and the topic of on-device data annotation is timely and important.

**Weaknesses:**

1.	The novelty seems limited as PWS has already been explored in centralized settings. The paper's main contribution appears to be the adaptation of an existing technique to a federated scenario rather than introducing a fundamentally new approach.

2.	Figure 1 aims to visualize the strategy for generating LFs in the WSHFL method. However, the figure appears to be a high-level representation without detailed annotations or explanations. The flow between the different stages (a to d) and the significance of the values associated with the "IF nice" statements are not immediately clear. A more detailed caption or accompanying text might help in understanding the figure's content.

3.	The paper mentions comparisons with fully supervised baselines, but there seems to be a lack of comprehensive comparison with state-of-the-art methods in FSSL. Such a comparison would be crucial to establish the proposed method's effectiveness and relevance in the current research landscape.

**Questions:**

see weaknesses.

---

> ### Author Response · Authors · 2023-11-16
> **Thanks for reviewing! Our rebuttal**
>
> Dear Reviewer W4Yv,
>
> We thank you for your time and effort in reviewing our paper. We are happy that you like our motivating example. As you point out, on-device data annotation is a timely and relevant challenge.
>
> We have commented on the concerns on presentation and novelty raised by the reviewers in our general response. Regarding your concern on the *lack of comprehensive comparison with state-of-the-art methods in FSSL*, we want to note that most of these techniques have been proposed for image data and rely on some form of weak or strong data augmentation [1, 3, 5, 7, 8, 9, 10, 14, 15]. Thus, it is not trivial to naively extend them to text and time-series data. As such FSSL for new data modalities is an exciting research direction in itself. Having said that, we are **actively working on modifying some of these techniques with the aim of addressing your concern in time**.
>
> Please let us know if there are any further areas we can improve on.

---

> > ### Author Response · Authors · 2023-11-21
> > **Summary of changes and a humble request for response**
> >
> > Dear Reviewer W4Yv,
> > Thank you for your time and effort in improving our paper.
> >
> > We wanted to draw your attention to the changes that we have made to the manuscript in response to your valuable comments and feedback. A summary of these changes is outlined in the general comment we posted.  **We would specifically like to draw your attention to our new experiments on federated semi-supervised learning that you emphasized (Appendix A.7).** In addition, we've incorporated changes to Figure 1 as per your suggestions, and introduced Appendix A.1, which complements Figure 1 with a detailed caption and explanation.
> >
> > *We are eagerly looking forward to your response. If our revisions have effectively addressed your concerns, we kindly request you to consider adjusting your scores accordingly.*
> >
> > Thank you once again for your thoughtful review.

---

### Official Review · Reviewer_mKax · 2023-11-04

**Soundness:** 2 fair
**Presentation:** 3 good
**Contribution:** 3 good
**Rating:** 6
**Confidence:** 4

**Summary:**

The paper discusses the challenges of leveraging on-device data for training intelligent mobile applications. The data is distributed on client devices and it is sensitive data, making it difficult to obtain expert annotations for traditional supervised machine learning. Current federated learning techniques typically use unsupervised approaches and cannot capture expert knowledge through data annotations. To address this issue, the paper introduces a method called Weak Supervision Heuristics for Federated Learning (WSHFL). WSHFL utilizes labeling functions, which are heuristic rules, to annotate on-device data in cross-device federated settings. The paper presents experimental results across two data modalities, text and time-series, demonstrating that WSHFL achieves competitive performance compared to fully supervised methods without the need for direct data annotations.

**Strengths:**

1. This paper addresses the problem of obtaining labels for the data distributed across devices and training a model using the labeling signal obtained. I like the motivation and introduction of the problem.


2. The authors provide a solution based on programmatic weak supervision (PWS) in which the expert feedback is incorporated using a set of heuristic rules, which is more efficient than asking for annotations of each data point. The authors apply it in the federated learning setting since the data cannot leave the client device i.e. not sharable. The proposed solution generates a set of candidate labeling functions on each client and then uses expert feedback to select accurate labeling functions that are used to train the model using WeaSEL model (Cachay et al. 2021).


3. Experiments on natural language and time series domains are provided that demonstrate the feasibility and effectiveness of the proposed approach WSHFL.

**Weaknesses:**

1. I find the method section 3 dense and confusing.  I have following confusions/questions. How are the candidate LFs generated on client? What is exactly happening at the server, in particular in ExpertQuery function? This function is providing $u_t$ and the TrainClient function is giving estimates $\hat{u}_t$. What is the neural network trained on and for what purpose? $u_j$ is a variable denoting whether $\lambda_j$ is selected or not, but statements like “ where we sequentially inspect the candidates in or der to discover members of our desired class ($u_j=1)$ make it look like it is one of the classes in $\mathcal{Y}$. The presentation of section 3 can be improved greatly to make it more lucid. There is very little horizontal space between the algorithm block.

2. The solution is aimed at federated learning setting, so privacy should be ensured. It is not clear to me whether LFs can leak personal information and does the overall solution ensures privacy.

**Questions:**

1. How does the method ensure privacy? The data is not shared with the experts/serve but I think there is a potential risk of information leak via labeling functions.
2. How many times are each method run in experiments? Does running more times reduce variance?

---

> ### Author Response · Authors · 2023-11-16
> **Thanks for reviewing! Our rebuttal**
>
> Dear Reviewer mKax,
>
> We thank you for your time and effort in reviewing our paper.
>
> We are glad that you found our work to be well motivated, feasible and effective. We are actively working on improving the readability of Section 3, and have commented on the privacy concerns raised by the reviewers in our general response. Below are answers to your specific questions:
>
> **How are the candidate LFs generated on the client?**
>
> Section 4 describes how LFs are generated in each client. Figures 2 and 3 show examples of text and time-series LFs respectively.
>
> 1. To generate text LFs, each client first identifies a set of unigrams (i.e. words) within a certain document frequency range. For example, say the client identifies 2 unigrams in their vocabulary: [nice, bad]. To create labeling functions, each client takes the cross product of these unigrams and the set of possible labels. If the set of possible labels is [negative sentiment, positive sentiment], then the client generates 4 LFs: [nice → positive sentiment, nice → negative sentiment, bad → positive sentiment, bad → negative sentiment]. To find keywords, we use the CountVectorizer implementation in scikit-learn for the same.
> 2. To generate time-series labeling functions, we first cluster time-series into $k$ clusters. We refer to the cluster representatives (the mean time-series) of these clusters as representative templates. To construct labeling functions, we take the cross product of these representative templates and the set of possible labels. The time-series labeling function is shown in Fig. 3. Intuitively, the closer a time-series is to a representative template, the more likely it is to be assigned to the class corresponding to the representative template.
>
> We have now added these toy examples to the appendix.
>
> **What is exactly happening at the server, in particular in `ExpertQuery` function?**
>
> `ExpertQuery` happens at the server, and is the process by which a human domain expert assigns $u_j=1$ (LF is useful/accurate) or $u_j=0$ (LF is not useful/accurate) to the given LF. In our work, the expert is positioned at the server.
>
> **What is the neural network trained on and for what purpose?**
>
> From the context of your question, we assume you are referring to the neural network $h_k$. This network is trained on the LFs that have already been inspected by the expert $h_k: \lambda_j \to u_j$ and, intuitively, predicts the probability that a new LF will be considered useful by the expert. We use this network to later on to select which LF we will actually show to the expert for inspection.
> How many times are each method run in experiments? Does running more times reduce variance? We mention in the "Methods and Models" section that "Unless mentioned otherwise, we repeat each experiment five times with different random seeds and report the mean and standard deviation." In preliminary analyses, we ran some experiments with 10 repetitions, but the behavior of the error bars did not qualitatively change.
>
> Please let us know if there are any further areas we can improve on. Based on your questions, we have already made some changes to improve the readability of our work.

---

> > ### Author Response · Authors · 2023-11-21
> > **Summary of changes and a humble request for response**
> >
> > Dear Reviewer mKax,
> >
> > Thank you for your time and effort in improving our paper.
> >
> > We wanted to draw your attention to the changes that we have made to the manuscript in response to your valuable comments and feedback. A summary of these changes is available in the general comment we have posted. We focused on enhancing the readability of our manuscript by updating Section 3, incorporating additional details and examples in the appendix, and engaging in a more comprehensive discussion on the potential privacy risks associated with labeling functions (LFs).
> >
> > *We are eagerly looking forward to your response. If our revisions have effectively addressed your concerns, we kindly request you to consider adjusting your scores accordingly.*
> >
> > Thank you once again for your thoughtful review.

---

> > ### Comment · Reviewer_mKax · 2023-11-23
> >
> > Appreciate your response to my questions. Most of my concerns are addressed. I would still encourage the authors to enhance the clarity of the paper in particular I like the conceptual contribution of the paper and I think the process of LF generation, role of human/expert in LF selection should be super clear early in the paper, if possible have a few more examples/illustrations.

---

### Author Response · Authors · 2023-11-16
**General Response (1/N)**

Dear Reviewers and Area Chair,

We sincerely appreciate your time and for offering valuable feedback.

We are glad that all reviewers found our work to be well motivated, and grounded in real-world impact through our experiments on ECG data. We are also happy that the reviewers appreciated our experiments on time-series data, which has received very little attention in federated learning (reviewer qXVo). Finally, we thank Reviewer W4Yv for pointing out how the needs for on-device expert annotation are not met by existing work.

Below we would like to highlight three general areas of improvement and discussion that all the reviewers have pointed out:

## Clarity and organization
Reviewers mKax and W4Yv pointed out areas of improvement in Section 3 and Figure 1. We are actively working on improving these elements to make them easier to follow. As of now, we have included more detail in Figure 1 (reviewer W4Yv), and we have consistently differentiated between labels $y$  and expert labels $u$ (reviewer mKax) in Section 3.

## Privacy leakage from unigram labeling functions
We agree with Reviewers qXVo and mKax that some families of LFs can leak private information, and already acknowledge these risks in the Societal Impacts section of our work.

Our methodology already follows the privacy principle of data minimization [16, 18] by choosing unigrams based on TF-IDF, therefore never exchanging infrequent words which can be indicative of individual clients. The risk of private information leakage could be further minimized by showing clusters of unigrams instead of individual keywords as text labeling functions, or we could derive unigrams from a vocabulary that is shared by the server to the clients. To further reduce privacy risks, we can leverage techniques such as SynTF [17] to create private term frequency vectors. Finally, under the appropriate threat model, we can also envision a mechanism in which the expert discards suspicious LFs, as the chosen LFs are inspected by the expert before they are sent to other clients.

We don’t make any claims about ensuring privacy in our work, as we don’t formally work with formalisms such as differential privacy. However, depending on the problem, it is possible to use simple interpretable classifiers as LFs, and use differential privacy in this setting.

We will add these discussions to the appendix in our future revision.

## Novelty
We would like to highlight the following **novel contributions** of our work to federated learning:

### Novel problem setting - supervising on-device data
To the best of our knowledge, there is no work which considers the problem of providing expert supervision to on-device data, when clients either do not have time or the expertise to label their own data. One solution would be to utilize techniques in federated semi- or self-supervised learning (FSSL), but these techniques are not directly applicable to our setting for the following reasons:

1. Studies considering the labels-at-client setting cannot be used because they assume that clients have partially labeled data. In the practical setting that we consider, clients either do not have the time or expertise to label their own data [2, 3, 4, 5, 8, 10, 14, 15].
2. Studies which consider the labels-at-server setting are not directly usable since they assume that the server has a limited amount of labeled data [6, 7, 8, 9, 10, 12, 13].
3. Finally, a few studies consider the self-supervised federated learning setting, where the goal is to learn unsupervised representations. These studies still require labeled data for classification [1, 11].
4. All of these studies are only evaluated on image data, and most of them rely on some form of weak or strong data augmentation techniques, which cannot be naively extended to text or time-series data.

### Weak supervision in a federated setting
While programmatic weak supervision has been studied in a centralized setting, extending it to the federated setting is not straightforward. We considered various nuances of this problem, such as how to generate candidate LFs at scale in a distributed manner, and how to properly account for the heterogeneity of the federated data when aggregating estimates $\hat{u}_j$. In our “Conclusions” section (Section 7), we also note several under-explored directions in this space, e.g., the investigation of privacy preserving LFs.

### Modeling time-series data
Time-series is a prevalent data modality in many domains such as healthcare, yet very few studies have considered the problem of modeling time-series in weakly supervised and federated settings.

We are making several changes to our manuscript. **All of these changes are in dark blue.**

---

> ### Author Response · Authors · 2023-11-16
> **General Response (2/N)**
>
> ## References
> [1] Makhija, Disha, Nhat Ho, and Joydeep Ghosh. "Federated self-supervised learning for heterogeneous clients." arXiv preprint arXiv:2205.12493 (2022).
>
> [2] Wang, Zhiguo, et al. "Federated semi-supervised learning with class distribution mismatch." arXiv preprint arXiv:2111.00010 (2021).
>
> [3] Fan, Chenyou, Junjie Hu, and Jianwei Huang. "Private semi-supervised federated learning." International Joint Conference on Artificial Intelligence. 2022.
>
> [4] Lin, Xinyang, et al. "Federated learning with positive and unlabeled data." International Conference on Machine Learning. PMLR, 2022.
>
> [5] Li, Simou, et al. "FedUTN: federated self-supervised learning with updating target network." Applied Intelligence 53.9 (2023): 10879-10892.
>
> [6] Zhang, Zhe, et al. "Semi-supervised federated learning with non-iid data: Algorithm and system design." 2021 IEEE 23rd Int Conf on High Performance Computing & Communications; 7th Int Conf on Data Science & Systems; 19th Int Conf on Smart City; 7th Int Conf on Dependability in Sensor, Cloud & Big Data Systems & Application (HPCC/DSS/SmartCity/DependSys). IEEE, 2021.
>
> [7] Diao, Enmao, Jie Ding, and Vahid Tarokh. "SemiFL: Semi-supervised federated learning for unlabeled clients with alternate training." Advances in Neural Information Processing Systems 35 (2022): 17871-17884.
>
> [8] Long, Zewei, et al. "FedSiam: Towards adaptive federated semi-supervised learning." arXiv preprint arXiv:2012.03292 (2020).
> [9] Bian, Jieming, Zhu Fu, and Jie Xu. "FedSEAL: Semi-supervised federated learning with self-ensemble learning and negative learning." arXiv preprint arXiv:2110.07829 (2021).
>
> [10] Liang, Xiaoxiao, et al. "Rscfed: Random sampling consensus federated semi-supervised learning." Proceedings of the IEEE/CVF Conference on Computer Vision and Pattern Recognition. 2022.
>
> [11] Zhuang, Weiming, Yonggang Wen, and Shuai Zhang. "Divergence-aware federated self-supervised learning." arXiv preprint arXiv:2204.04385 (2022).
>
> [12] Zhang, Zhengming, et al. "Improving semi-supervised federated learning by reducing the gradient diversity of models." 2021 IEEE International Conference on Big Data (Big Data). IEEE, 2021.
>
> [13] Diao, Enmao, Jie Ding, and Vahid Tarokh. "SemiFL: Communication efficient semi-supervised federated learning with unlabeled clients." (2021).
>
> [14] Lin, Haowen, et al. "Semifed: Semi-supervised federated learning with consistency and pseudo-labeling." arXiv preprint arXiv:2108.09412 (2021).
>
> [15] Yan, Rui, et al. "Label-efficient self-supervised federated learning for tackling data heterogeneity in medical imaging." IEEE Transactions on Medical Imaging (2023).
>
> [16] Kairouz, P., McMahan, H. B., Avent, B., Bellet, A., Bennis, M., Bhagoji, A. N., ... & Zhao, S. (2021). Advances and open problems in federated learning. Foundations and Trends in Machine Learning, 14(1–2), 1-210.
>
> [17] Weggenmann, Benjamin, and Florian Kerschbaum. "Syntf: Synthetic and differentially private term frequency vectors for privacy-preserving text mining." The 41st International ACM SIGIR Conference on Research & Development in Information Retrieval. 2018.
>
> [18] Bonawitz, Kallista, et al. "Federated learning and privacy." Communications of the ACM 65.4 (2022): 90-97.
>
> [19] Li, Jeffrey, et al. "Differentially Private Meta-Learning." International Conference on Learning Representations. 2019.

---

### Author Response · Authors · 2023-11-21
**Summary of Changes**

Dear Reviewers and Area Chair,

We would like to express our sincere gratitude for your time and feedback. We have made several changes over the past few days to improve the clarity and experimentation as pointed out by all the reviewers.

Below we provide a summary of changes. All of these *changes are in dark blue*.

### Comparison with federated semi-supervised learning
Based on the suggestions by Reviewer W4Yv, we compared our method (WSHFL) with two state-of-the-art federated semi-supervised learning techniques with labels at the server: SemiFL [1] and FRGD [2], on the IMDb dataset. **We found that WSHFL outperforms the compared semi-supervised learning methods across 5 independent runs on the IMDb dataset (Fig. 16 on Page 26 of the Appendix). The detailed experiment setup is described in Appendix A.7.**

We also provided a deeper discussion of federated semi- and self-supervised learning methods, and their limitations in the context of the problem that we are trying to solve. You can find this in Appendix A.7. In addition, we would like to note that ours are the first set of reproducible federated experiments on non-image data in settings where the clients hold unlabeled data.

### Clarity and organization
In response to Reviewer mKax’s questions and suggestions, we have made changes to Section 3. We also added a table of symbols and equations in Appendix A.10 to serve as a convenient point of reference for readers, enhancing the overall clarity of our manuscript.

In addition, we added annotations to Figure 1 in response to Reviewer W4Yv’s comments. We have supplemented a copy of Figure 1 (Fig. 9 in Appendix A.1, Page 16) with a comprehensive caption to provide a detailed explanation of WSHFL’s approach to supervising on-device data.

### Reproducible science
We thank Reviewer qXVo for recognizing the contributions of our work and offering valuable suggestions to enhance its impact, particularly by advocating for the release of our code as open source.  In response, we promptly made our code accessible at https://anonymous.4open.science/r/wshfl_pipeline-A13C/. We anticipate that sharing our code will encourage further research on the significant challenge of learning from unlabeled on-device data in non-image data modalities.

### Privacy leakage from unigram labeling functions
We improved the clarity of the Societal Impacts section of our manuscript (under Section 7) to reflect the deeper discussions we have had with reviewers mKax and qXVo on the risk of potential information leakage from some families of LFs.

**We hope that our responses and efforts have adequately addressed the reviewers’ concerns and questions. Please let us know if you have any remaining questions or concerns. We are looking forward to your responses.**

### References
[1] Diao, Enmao, Jie Ding, and Vahid Tarokh. "SemiFL: Semi-supervised federated learning for unlabeled clients with alternate training." Advances in Neural Information Processing Systems 35 (2022): 17871-17884.

[2] Zhang, Zhengming, et al. "Improving semi-supervised federated learning by reducing the gradient diversity of models." 2021 IEEE International Conference on Big Data (Big Data). IEEE, 2021.

---

### Meta-Review · Area_Chair_z3vm · 2023-12-12

**Metareview:**

After careful consideration of the reviews and the authors' responses, it is clear that while the authors have made efforts to address concerns around clarity, organization, and privacy risks in their WSHFL methodology, the paper still lacks the necessary impact required for acceptance. The reviewers brought up critical issues regarding the novelty of the key ideas, with the adaptation of programmatic weak supervision to a federated setting being more incremental than revolutionary. Moreover, concerns about potential privacy leakage through the labeling functions, despite efforts to address them, remain a significant concern in the context of the federated learning paradigm, which prioritizes data privacy.

The experiments, although conducted across different datasets, fail to compare against a broader range of state-of-the-art federated semi-supervised learning techniques, which leaves the actual impact and performance gain of WSHFL in the field unclear. The readability of the manuscript has improved, according to the reviewers, but there is room for enhancement, particularly in the explanations around the generation of labeling functions and the role of the expert in LF selection.

Overall, despite the authors’ response to the reviews, the paper has not met the threshold for acceptance owing to lingering concerns about originality and depth. It is recommended that the authors further refine their contribution, strengthen the privacy-preserving aspects of their methodology, and provide more extensive comparisons with state-of-the-art methods to establish the significance of their work in future submissions.

**Justification For Why Not Higher Score:**

The experiments, although conducted across different datasets, fail to compare against a broader range of state-of-the-art federated semi-supervised learning techniques, which leaves the actual impact and performance gain of WSHFL in the field unclear. The readability of the manuscript has improved, according to the reviewers, but there is room for enhancement, particularly in the explanations around the generation of labeling functions and the role of the expert in LF selection.

**Justification For Why Not Lower Score:**

N/A

---

### Decision · Program_Chairs · 2024-01-16

Reject